# Genetically incorporated crosslinkers reveal NleE attenuates host autophagy dependent on PSMD10

**Jingxiang Li[1], Shupan Guo[1], Fangni Chai[1], Qi Sun[1], Pan Li[1], Li Gao[2], Lunzhi Dai[2], Xiaoxiao Ouyang[1], Zhihui Zhou[1], Li Zhou[1], Wei Cheng[1], Shiqian Qi[1], Kefeng Lu[1], Haiyan Ren[1]\***

[1]Division of Respiratory and Critical Care Medicine, State Key Laboratory of Biotherapy, West China Hospital of Sichuan University and Collaborative Innovation Center of Biotherapy, Chengdu, China; [2]Department of General Practice and National Clinical Research Center for Geriatrics, State Key Laboratory of Biotherapy, West China Hospital, and Sichuan University, Chengdu, China

**Abstract** Autophagy acts as a pivotal innate immune response against infection. Some virulence effectors subvert the host autophagic machinery to escape the surveillance of autophagy. The mechanism by which pathogens interact with host autophagy remains mostly unclear. However, traditional strategies often have difficulty identifying host proteins that interact with effectors due to the weak, dynamic, and transient nature of these interactions. Here, we found that Enteropathogenic *Escherichia coli* (EPEC) regulates autophagosome formation in host cells dependent on effector NleE. The 26S Proteasome Regulatory Subunit 10 (PSMD10) was identified as a direct interaction partner of NleE in living cells by employing genetically incorporated crosslinkers. Pairwise chemical crosslinking revealed that NleE interacts with the N-terminus of PSMD10. We demonstrated that PSMD10 homodimerization is necessary for its interaction with ATG7 and promotion of autophagy, but not necessary for PSMD10 interaction with ATG12. Therefore, NleE-mediated PSMD10 in monomeric state attenuates host autophagosome formation. Our study reveals the mechanism through which EPEC attenuates host autophagy activity.

**\*For correspondence:**
hyren@scu.edu.cn

## Introduction

Autophagy, a process referring to engulfment of a portion of the cytosol in a double-membrane autophagosome to lysosomes for degradation, plays a vital role in the host response to pathogens. Antibacterial autophagy, known as xenophagy, delivers intracellular bacteria to lysosomes for degradation (*Xu et al., 2019*; *Choi et al., 2013*). Autophagy not only plays a role in pathogen sensing and restriction, but also implicates in many other immune processes, such as proper immune cell differentiation, regulation of pattern recognition receptors, cytokine production, inflammasome activation, antigen presentation, and lymphocyte homeostasis, which contribute to host response to pathogens (*Pareja and Colombo, 2013*; *Germic et al., 2019*; *Gomes and Dikic, 2014*). However, various bacterial pathogens have evolved strategies to combat the autophagy pathway in order to enhance their survival (*Agarwal et al., 2015*; *He et al., 2017*; *Gannagé et al., 2009*). These strategies include escape from autophagy recognition, suppression of autophagy initiation, inhibition of autophagosome formation, blockade of autophagosome–lysosome fusion, hijacking of autophagy for replication, etc. (*Wu and Li, 2019*). Considering the variety of mechanisms utilized by different pathogens, understanding the mechanism by which each microbe manipulates autophagy is key to developing effective strategies for clinical treatment.

Whether and how pathogens affect autophagy in host cells remains largely elusive. Enteropathogenic *Escherichia coli* (EPEC) species are gram-negative extracellular pathogens that mainly infect the human intestine, causing life-threatening diarrhea in immunocompromised individuals. We found that EPEC partially suppresses autophagosome formation in host cells in a NleE dependent manner. NleE was the first identified member of a class of S-adenosyl-L-methionine (SAM)-dependent methyltransferases in EPEC. To investigate how NleE interferes with autophagy, we sought to identify NleE interaction partners in host cells. Traditional strategies, such as immunoprecipitation and affinity purification, often have difficulty identifying host proteins that interact with effectors due to the weak, dynamic and transient nature of these interactions in living cells. Covalent capture enables the identification of weak and transient interactions with enhanced specificity, reliability, and accuracy (*Yu and Huang, 2018*; *Preston and Wilson, 2013*; *Sinz, 2018*). In particular, genetically encoded crosslinking technology has emerged as an attractive strategy for the investigation of native protein–protein interactions in living cells (*Coin, 2018*; *Coin et al., 2013*; *Yang et al., 2016*; *Zhang et al., 2011*; *Zhang et al., 2017*; *Tang et al., 2018*). However, this technology has been applied mostly to map peptide–protein and protein–protein interactions. The low yield of photocrosslinking and the high background signals of crosslinked samples in mass spectrometry (MS) analysis significantly hinder this technique for identification of new interaction partners, especially in mammalian cells, which usually contain small amounts of proteins (*Coin, 2018*). We optimized the process and successfully identified PSMD10 as an interaction partner of NleE in mammalian cells using genetically incorporated unnatural amino acids (Uaas).

The interaction mechanism between NleE and PSMD10 was revealed by pairwise chemical crosslinking. Traditionally, disulfide bond formation between specific Cys residues has been used to detect intermolecular proximity in vivo (*Dong et al., 2012*). However, the sensitivity of disulfide bonds to reducing environments limits its application. Pairwise chemical crosslinking arises from the reactions that occur between genetically incorporated proximity-enabled Uaas and specific amino acids (such as Cys and Lys) when the two groups are proximal to each other (*Coin et al., 2013*; *Xiang et al., 2013*; *Xiang et al., 2014*; *Ren, 2020*; *Wang, 2017*). The high specificity and stability of crosslinking make it ideal for detecting the intermolecular proximities of proteins in living cells (*Yang et al., 2017*). In addition, proximity-enabled crosslinking likely depicts dynamic interaction conformations under physiological conditions.

Our study reveals a new mechanism by which pathogens attenuate the host defense response. NleE regulates autophagosome formation in a PSMD10 dependent manner in host cells. Further studies reveal that homodimerization of PSMD10 is critical for its interaction with ATG7 and ATG10, but not for its interaction with ATG12. Specifically, NleE interacts with the N-terminus of PSMD10 and suppresses PSMD10 homodimerization.

## Results

### NleE attenuates autophagosome formation in host cells

Gram-negative bacterial cell wall-derived lipopolysaccharide (LPS) is a potent proinflammatory pathogen-associated molecule. LPS treatment significantly induced autophagy in HeLa cells, as measured by increases in the numbers of LC3-positive puncta and the levels of lipidated LC3 (LC3-II) (*Figure 1A,B*; *Lapaquette et al., 2010*). We found that LPS-induced autophagy was partially suppressed by the T3SS effector NleE (*Figure 1A,B*). The NleE-mediated suppression phenotype was also observed in starvation-induced autophagy and basel-level autophagy (*Figure 1C,D*). We next examined whether NleE affects another LC3-mediated process. Mammalian cells were transfected with polystyrene beads to mimic bacterial infection, and LC3 signaling was efficiently induced around the beads (*Figure 1E*; *Xu et al., 2019*; *Kobayashi et al., 2010*). Beads entrapment by LC3-positive compartments were partially suppressed by NleE (*Figure 1E*). Furthermore, live EPEC, but not EPEC ΔNleE, attenuated autophagy in macrophages derived from THP-1 cells, as indicated by decreased LC3-positive puncta (*Figure 1F*).

To determine which stage of autophagy was affected by NleE, we examined the mCherry-GFP-LC3 reporter (*Figure 1G*). The GFP signal of the mCherry-GFP-LC3 reporter is quenched in acidified compartments. The red signal represents functional autolysosomes (AL), while yellow puncta indicate autophagic structures (AP) prior to the formation of acidified AP such as phagophores

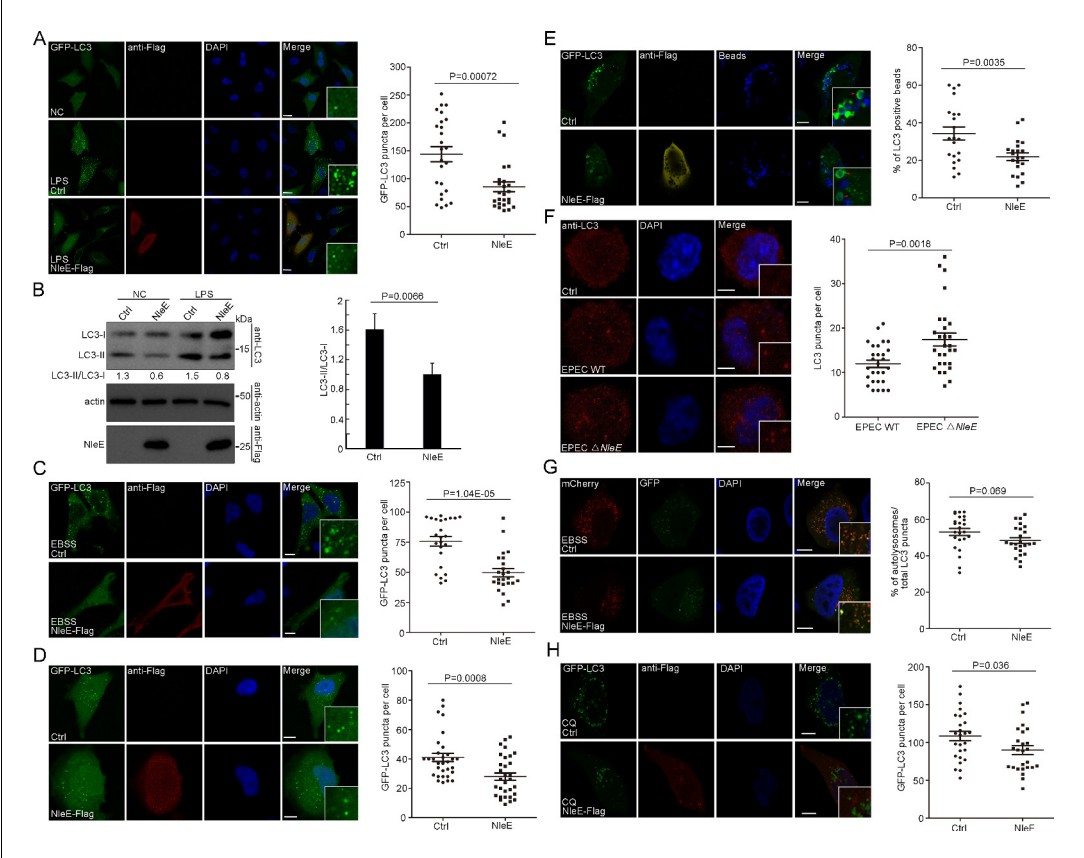

**Figure 1.** NleE attenuates autophagosome formation in host cells. (**A, B**) NleE attenuates LPS-induced autophagy in HeLa cells. (**A**) HeLa cells expressing GFP-LC3 were transiently transfected with NleE-Flag. Representative images are shown. Scale bars, 20 μm. GFP-LC3 puncta per cell were quantified (N = 25). (**B**) Immunoblotting analysis of endogenous LC3 lipidation (LC3-II) in HeLa cells. Shown are the LC3–I (cytosolic) and LC3–II (lipid conjugated) forms as detected with antibodies to LC3. Experiments were repeated three times and quantified. (**C, D**) Starvation-induced autophagy and basel-level autophagy were partially suppressed by NleE in HeLa cells. Representative images are shown for each condition. Scale bars, 20 μm. GFP-LC3 puncta per cell were quantified (N = 24 for C and N ≥ 30 for D). (**E**) The LC3-mediated process was partially suppressed by NleE in HeLa cells. Arrowheads indicate internalized beads entrapped by LC3. Scale bars, 10 μm. The percentages of LC3 positive beads per cell were quantified (N = 21). (**F**) Live EPEC attenuates host autophagy dependent on NleE. WT EPEC and EPEC ΔNleE were cocultured with THP-1-induced macrophages. Cells were stained with anti-LC3 antibody. Scale bars, 5 μm. LC3 puncta per cell were quantified (N = 28). (**G–H**) NleE interferes with the autophagosome formation, but not the autophagosome fusion step. (**G**) HeLa cells were transfected with mCherry-GFP-LC3 or mCherry-GFP-LC3 and NleE. Representative images are shown. Scale bars, 10 μm. The ratio of autolysosomes (red) to the total autophagic vesicles (red+yellow) was quantified (N = 24). (**H**) NleE attenuates autophagy in choroquine (CQ)-treated host cells. Scale bars, 10 μm. GFP-LC3 puncta per cell were quantified (N = 26). All puncta quantification was performed in cells from three independent experiments. All quantification data represent the mean ± SD of three independent experiments.

The online version of this article includes the following source data and figure supplement(s) for figure 1:

**Source data 1.** Numerical data for *Figure 1A*.
**Source data 2.** Numerical data for *Figure 1B*.
**Source data 3.** Numerical data for *Figure 1C*.
**Source data 4.** Numerical data for *Figure 1D*.
**Source data 5.** Numerical data for *Figure 1E*.
**Source data 6.** Numerical data for *Figure 1F*.
**Source data 7.** Numerical data for *Figure 1G*.
**Source data 8.** Numerical data for *Figure 1H*.
**Source data 9.** Original western blot files for *Figure 1*.
**Figure supplement 1.** NleE inhibition of autophagy affects IL-6 production toward EPEC.
**Figure supplement 1—source data 1.** Numerical data for *Figure 1—figure supplement 1A*.
**Figure supplement 1—source data 2.** Numerical data for *Figure 1—figure supplement 1B*.
**Figure supplement 1—source data 3.** Numerical data for *Figure 1—figure supplement 1C*.

(autophagosomal precursors), nascent autophagosomes, and unacidified amphisomes (fused vesicles between autophagosomes and endocytic vesicles). The total number of autophagic vesicles (sum of AL + AP) decreased in cells expressing NleE (*Figure 1G*). However, the ratio of AL to the total autophagic vesicles was not affected by NleE, indicating that NleE interferes with autophagosome formation (*Figure 1G*). Consistently, GFP-LC3 puncta also decreased in NleE-expressing cells treated with the lysosome inhibitor chloroquine (CQ), which further indicated that NleE functions in inhibiting autophagosome formation (*Figure 1H*). Collectively, NleE blocks autophagosome formation in host cells.

To test whether NleE inhibition of autophagy affects host responses toward EPEC, we examined IL-6 production in host cells. IL-6 production in macrophages increased during EPEC ΔNleE infection, compared with WT EPEC infection (*Figure 1—figure supplement 1A*). Since NleE suppressed NF-κB pathway which is also involved in cytokine regulation (*Zhang et al., 2012*; *Zhang et al., 2008*), we constitutively actived the NF-κB pathway by expressing IKKb$^{CA}$ (S177E/S181E) in macrophages. NleE still suppressed IL-6 production during EPEC infection in macrophages expressing IKKb$^{CA}$, which indicated that NleE affects IL-6 production independent of TAB2/TAB3-mediated NF-κB suppression (*Figure 1—figure supplement 1B*). Furthermore, autophagy inhibitor treatment partially suppressed IL-6 production of macrophages during EPEC infection (*Figure 1—figure supplement 1C*).

## Genetically incorporated Uaas identified PSMD10 as an interaction partner of NleE

To explore the molecular mechanism by which NleE suppresses host autophagy, we sought to identify potential NleE interaction partners in living cells. Using the Tyrosine tRNA synthetase pair derived from *E. coli*, the photocrosslinking Uaa p-azido-phenylalanine (Azi, *Figure 2A*) was genetically incorporated into specific positions (X) of NleE-Flag. The expression level of NleE-Azi protein was ~50–100% of wild-type (WT) protein at 90% positions (*Figure 2—figure supplement 1A*). The reactive Uaas form crosslinking moieties upon UV irradiation and covalently capture proximal natural amino acids of the interaction partners only if the X$_{Uaa}$ is located at the protein interaction surface (*Figure 2B*). Azi was incorporated into NleE at 63 residues located in the substrate binding domain and the active center (*Figure 2—figure supplement 1B*). The covalent complexes were found to correspond to the adduct molecular weight (MW) on denaturing SDS–PAGE gels (*Figure 2B*). The most robust ultraviolet (UV)-dependent adduct band (~55 kDa) was detected in cells expressing NleE-K219Azi-Flag (*Figure 2C*). The covalent capture was site-specific, as we failed to detect the adduct band when Azi was introduced into positions near K219, such as N218 (*Figure 2C*). No crosslinking bands were found in the WT NleE samples and non-UV-treated NleE-K219Azi-Flag-negative controls (*Figure 2D*). Moreover, incorporation of another photocrosslinking Uaa (DiZPK, *Figure 2A*) at K219 of NleE also resulted in capture of covalent complexes of the same size (*Figure 2—figure supplement 1C*). Together, we found that the potential partners of NleE could be crosslinked by NleE-K219Uaa-Flag in live cells.

Next, the crosslinked NleE-K219Azi-Flag-His complexes were purified with anti-Flag beads, followed by nickel beads purification under denaturing condition to eliminate most of the noncovalent binding partners (*Figure 2—figure supplement 2A*). MS was used and identified 18 candidates (referred to as Y) that were present in NleE-K219Azi samples but absent or present at lower level in WT NleE samples (*Figure 2—figure supplement 2B*). All HA-tagged candidates were individually coexpressed with NleE-K219Azi-Flag in living cells for further screening. A clear crosslinking band was detected only in UV-treated NleE-K219Azi and PSMD10-HA coexpressing sample during anti-HA immunoblotting (*Figure 2—figure supplement 2C*). No other bands was detected in WT and non-UV-treated NleE-K219Azi controls, suggesting that PSMD10 crosslinked with NleE (*Figure 2E*). Western blots using anti-PSMD10 antibody also detected the covalent complex bands (*Figure 2E*). Fusion of NleE to GFP resulted in upshifting of the crosslink band to ~80 kDa on SDS–PAGE gels (*Figure 2—figure supplement 2D*). Although NleE is present in the cytoplasm and nucleus, we detected NleE and PSMD10 crosslinking only in the cytoplasm (*Figure 2—figure supplement 2E*). The NleE$_{△209IDSYMK214}$ mutant with Azi incorporated did not crosslink with PSMD10 (*Figure 2—figure supplement 2F*). The NleE$_{R107A}$ mutant that loss the SAM binding ability and Nle$_{E49AAAA52}$ mutant that lack 49GITR52 substrate binding motif also failed to crosslink with PSMD10 (*Figure 2—*

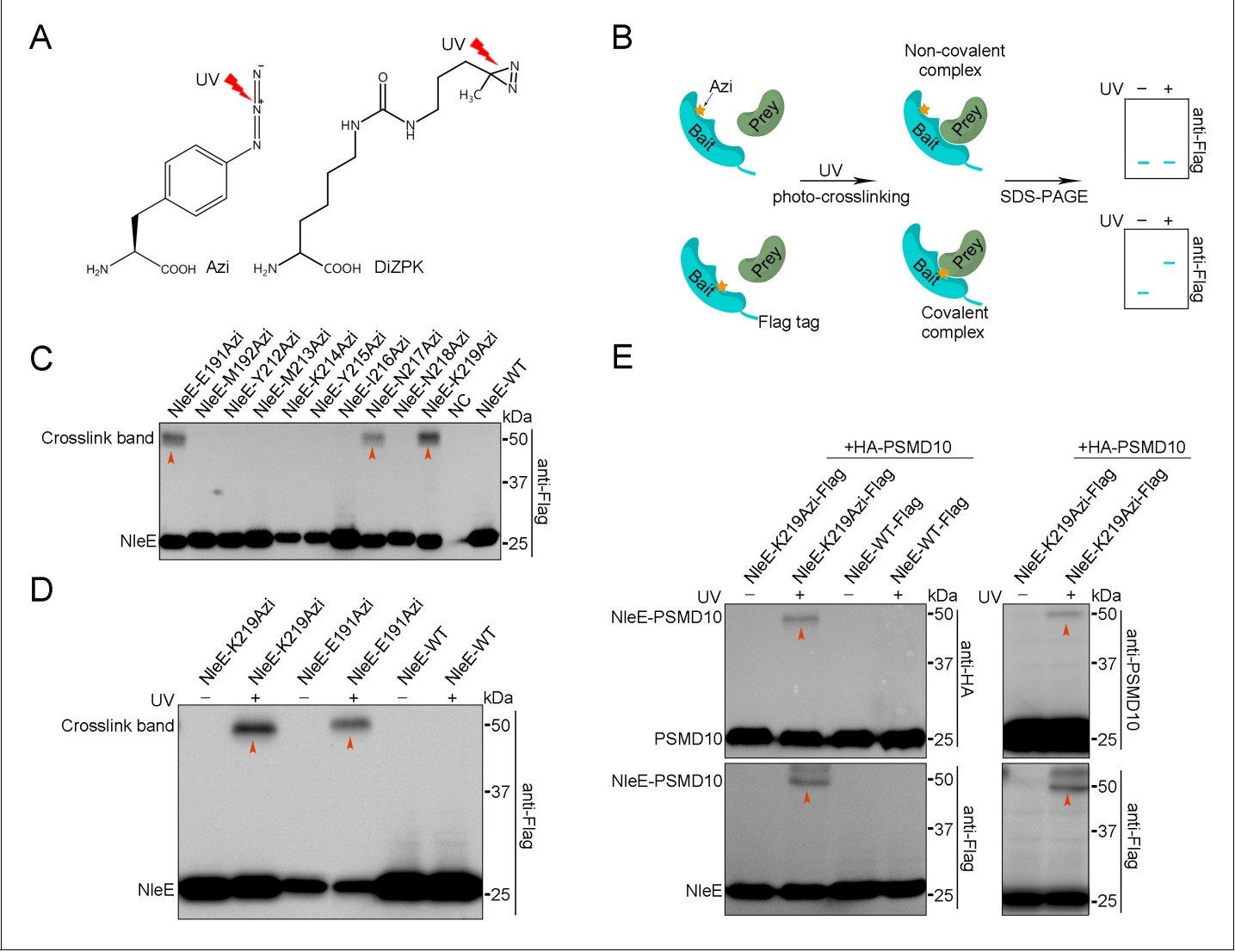

**Figure 2.** PSMD10 is an interaction partner of NleE in living cells. (A) Photocrosslinking Uaas used in this study. (B) Schematic diagram of genetically incorporated Uaas covalent capture interaction partners. (C) Photocrosslinking experiments map of NleE sites involved in covalent capture of interaction partners. Residues replaced by Azi are indicated in the upper row. Cell lysates were separated on SDS-PAGE gels and analyzed by immunoblotting using anti-Flag antibodies. (D) The photocrosslinking is Uaas and UV dependent. No crosslink band was identified in the WT NleE sample and NleE-K219Azi sample without UV treatment. (E) Validation of PSMD10 as a NleE interaction partner in living cells. Covalent complexes were detected with antibodies to HA, PSMD10, and Flag.

The online version of this article includes the following source data and figure supplement(s) for figure 2:

**Source data 1.** Original western blot files for *Figure 2*.
**Figure supplement 1.** Uaas-dependent capture of NleE covalent complexes.
**Figure supplement 1—source data 1.** Numerical data for *Figure 2—figure supplement 1A*.
**Figure supplement 1—source data 2.** Original western blot files for *Figure 2—figure supplement 1*.
**Figure supplement 2.** Identification of PSMD10 as a NleE interaction partner.
**Figure supplement 2—source data 1.** Numerical data for *Figure 2—figure supplement 2B*.
**Figure supplement 2—source data 2.** Original western blot files for *Figure 2—figure supplement 2*.

*figure supplement 2G*). These results indicated that the interaction between NleE and PSMD10 is specific. Thus, the PSMD10 protein interacts with NleE in living cells.

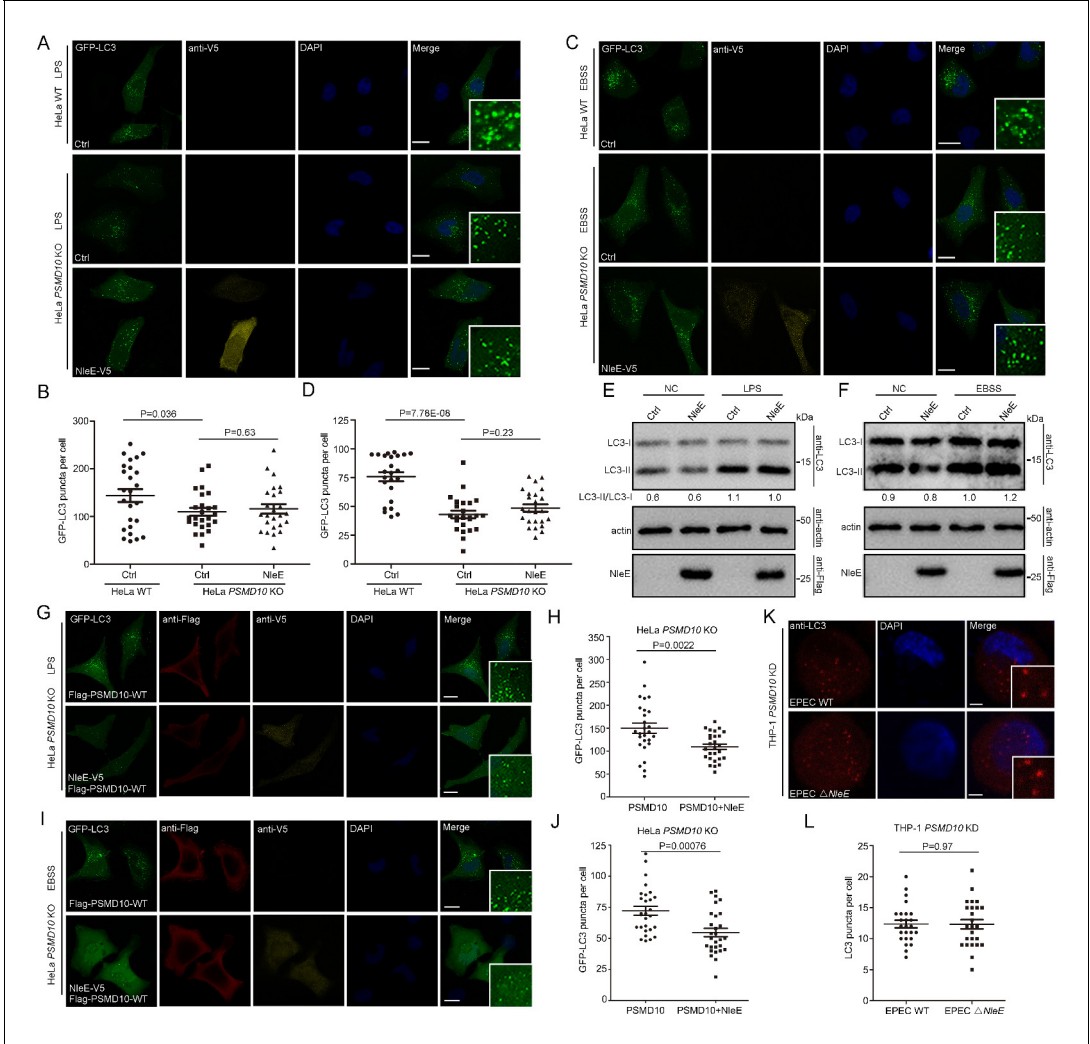

**Figure 3.** NleE regulates host autophagy dependent on PSMD10. (A–D) NleE fails to suppress host autophagy in *PSMD10 KO* cells. Representative images are shown. Scale bars, 20 μm. GFP-LC3 puncta per cell were quantified for LPS (B) and starvation-induced conditions (D) (N ≥ 24). (E, F) NleE does not affect the endogenous LC3-II/I ratio in *PSMD10 KO* cells. Shown are the LC3–I (cytosolic) and LC3–II (lipid conjugated) forms detected with anti-LC3 antibodies. Experiments were repeated three times. (G–J) WT PSMD10 restores the NleE suppression of autophagy in *PSMD10 KO* cells. Representative images are shown. Scale bars, 20 μm. LC3 puncta per cell were quantified for LPS (H) and starvation-induced conditions (J) (N≥27). (K, L) EPEC suppresses autophagy in a PSMD10 dependent manner. *PSMD10* KD THP-1 cells were induced with 200 ng/ml PMA for 48 hr and infected with EPEC WT or EPEC ΔNleE for 3 hr. Representative images are shown. Scale bars, 5 μm. (L) LC3 puncta per cell were quantified (N=25). All quantification was performed in cells from three independent experiments. All quantification data represent the mean ± SD of three independent experiments.

The online version of this article includes the following source data and figure supplement(s) for figure 3:

**Source data 1.** Numerical data for *Figure 3B*.
**Source data 2.** Numerical data for *Figure 3D*.
**Source data 3.** Numerical data for *Figure 3H*.
**Source data 4.** Numerical data for *Figure 3J*.
**Source data 5.** Numerical data for *Figure 3L*.
**Source data 6.** Original western blot files for *Figure 3*.
**Figure supplement 1.** NleE expression was not affected by PSMD10.
**Figure supplement 1—source data 1.** Numerical data for *Figure 3—figure supplement 1A*.
**Figure supplement 1—source data 2.** Original western blot files for *Figure 3—figure supplement 1*.

## PSMD10 is essential for NleE suppression of host autophagy

PSMD10 is a non-ATPase subunit of the 26S proteasome. Given that PSMD10 usually functions as a chaperone and interacts with target proteins to promote their degradation by the proteasome (*Lu et al., 2017*; *Hori et al., 1998*; *Dawson et al., 2002*), we evaluated the effect of PSMD10 on NleE protein degradation. We did not find obvious differences in NleE protein levels and ubiquitination levels in WT and *PSMD10 KO* cells (*Figure 3—figure supplement 1A*), suggesting that PSMD10 may not be involved in regulating NleE protein levels. Consistent with this finding, immunoprecipitation and pulldown experiments showed that PSMD10 interaction with 26S proteasome AAA-ATPase subunit Rpt3 was not affected by NleE (*Figure 3—figure supplement 1B–C*).

To determine whether the interaction of PSMD10 with NleE plays a role in regulating autophagy, we examined the effect of NleE on starvation and LPS-induced autophagy in *PSMD10*-deficient cells. Both LPS and starvation-induced autophagy reduced in *PSMD10 KO* cells as indicated by the number of GFP-LC3 puncta (*Figure 3A–D*). While NleE failed to further suppress autophagy in LPS-treated (*Figure 3A,B*) and starved-treated *PSMD10 KO* cells (*Figure 3C,D*). Consistently, the LC3-II/I level in *PSMD10 KO* cells was not altered by NleE (*Figure 3E,F*). Expression of full-length PSMD10 restored the autophagy suppression phenotype caused by NleE in LPS-treated (*Figure 3G,H*) and starved *PSMD10*-deficient cells (*Figure 3I,J*). In addition, EPEC infection failed to suppress autophagy in the PSMD10 KD macrophages (*Figure 3K*). Thus, the bacterial effector NleE regulates starvation and LPS-induced autophagy in host cells in a PSMD10-dependent manner.

## NleE interacts with the N-terminus of PSMD10

NleE was the first identified member of a class of SAM-dependent methyltransferases in EPEC. The methylated NleE substrate TAB2 showed a size shift in native gel (*Zhang et al., 2012*). However, NleE treatment did not result in a PSMD10 size shift in native gel, suggesting that PSMD10 was not methylated (*Figure 4—figure supplement 1A*). MS analyses failed to identify methylated Cys residues on PSMD10 either in living cells or in vitro, regardless the presence of SAM (*Figure 4—figure supplement 1B,C*). NleE-K219Azi crosslinking with PSMD10 was also not affected by SAM (*Figure 4—figure supplement 1D*). These results indicated that PSMD10 maybe not a substrate of NleE. Although the methylation activity of NleE may not necessary for its effect on PSMD10, the related mutations may cause slight changes of NleE structure and affect its crosslinking with PSMD10.

We next investigated the intermolecular interactions of residue pairs of NleE-PSMD10 complexes in living cells using proximity-enabled crosslinking Uaas. The proximity-enabled Uaa BetY, which selectively reacts with Cys, was incorporated into several sites of NleE individually (*Figure 4—figure supplement 2A*, *Figure 4A*; *Xiang et al., 2013*; *Xiang et al., 2014*; *Chen et al., 2014*; *Wang et al., 2018*; *Cigler et al., 2017*; *Xuan et al., 2017*). These NleE-XBetY mutants were coexpressed with wild-type (WT) PSMD10, which contains five Cys. NleE-XBetY-PSMD10 crosslinking bands were identified at multiple sites, all of which were located at the entrance of the active center (*Figure 4B,C*). Mutation of PSMD10 Cys residues that reacted with NleE-BetY prevented the formation of the covalent bond, allowing determination of the interacting amino acid pairs in the NleE-PSMD10 complexes (*Figure 4—figure supplement 2B*). We individually mutated all five Cys residues (C4, C11, C48, C107, and C180) in PSMD10. Photocrosslinking experiments showed that all PSMD10 Cys mutants retained NleE binding activity (*Figure 4—figure supplement 2C*). However, the C4S mutation in PSMD10 disrupted BetY-mediated covalent interaction between PSMD10 and NleE (*Figure 4D*). In summary, these results indicate that NleE interacts with the N-terminus of PSMD10.

## NleE blocks PSMD10 homodimerization

We found that purified PSMD10 was separated into two peaks by size-exclusion chromatography (SEC) (*Figure 5A*). Based on the multiangle laser light scattering analysis, the two peaks corresponded to the PSMD10 dimer and PSMD10 monomer (*Figure 5A*). The dimer fraction was analyzed on a denaturing gel and found to have an MW corresponding to that of the PSMD10 monomer (*Figure 5A*). PSMD10 homodimerization was also detected in live mammalian cells, as revealed by native gel electrophoresis and capture by multiple PSMD10-X-Azi mutants (*Figure 5B*, *Figure 5—figure supplement 1A*). Furthermore, both in living cells and in vitro, the PSMD10 homodimer

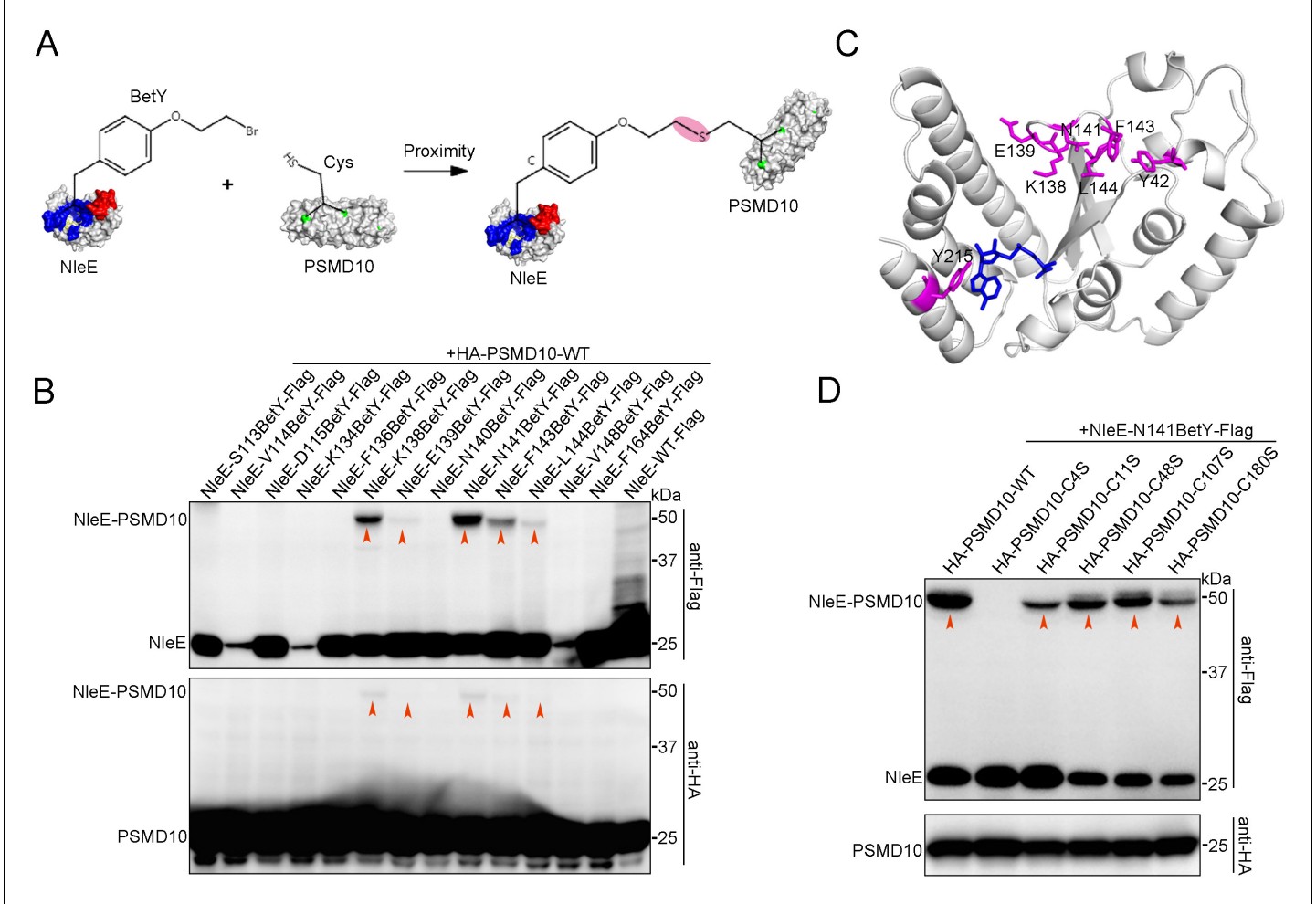

**Figure 4.** NleE interacts with the N-terminus of PSMD10. (**A**) Covalent crosslinking between cysteine (Cys) in PSMD10 and Uaa (BetY) in NleE. BetY was genetically incorporated in the substrate binding domain (colored red) and active center (colored blue) of NleE. (**B**) NleE-XBetY mutants covalent capture PSMD10. Residues X replaced by BetY are indicated in the upper row. (**C**) NleE structure with BetY positions crosslinked with PSMD10 colored in magenta. The NleE structure was downloaded from the Protein Data Bank with the accession code 4R29. (**D**) Cys4 mutation of PSMD10 disrupts covalent crosslinking with NleE-N141BetY. The mutated Cys in PSMD10 are indicated in the upper row.

The online version of this article includes the following source data and figure supplement(s) for figure 4:

**Source data 1.** Original western blot files for *Figure 4*.
**Figure supplement 1.** PSMD10 is not a NleE substrate.
**Figure supplement 1—source data 1.** Numerical data for *Figure 4—figure supplement 1B*.
**Figure supplement 1—source data 2.** Numerical data for *Figure 4—figure supplement 1C*.
**Figure supplement 1—source data 3.** Original western blot files for *Figure 4—figure supplement 1*.
**Figure supplement 2.** Pinpoint interaction residue pairs in NleE-PSMD10 complexes.
**Figure supplement 2—source data 1.** Numerical data for *Figure 4—figure supplement 2A*.
**Figure supplement 2—source data 2.** Original western blot files for *Figure 4—figure supplement 2*.

disappeared in the presence of reducing agents (DTT and β-mercaptoethanol [β-Me]), indicating that disulfide bond-mediated dimerization (*Figure 5C*, *Figure 5—figure supplement 1A*).

To identify the Cys residues in PSMD10 involved in dimerization, we mutated all five Cys residues individually. We failed to detect PSMD10 C4S homodimers both in vitro and in living cells on native gels (*Figure 5D*, *Figure 5—figure supplement 1B*). Consistently, the purified PSMD10 C4S protein resulted in only one peak corresponding to the PSMD10 monomer in SEC (*Figure 5A*). However, crosslinking with disuccinimidyl suberate (DSS) and genetically incorporated Azi allowed capture of homodimerized PSMD10 C4S mutants (*Figure 5—figure supplement 1C,D*), which indicate that

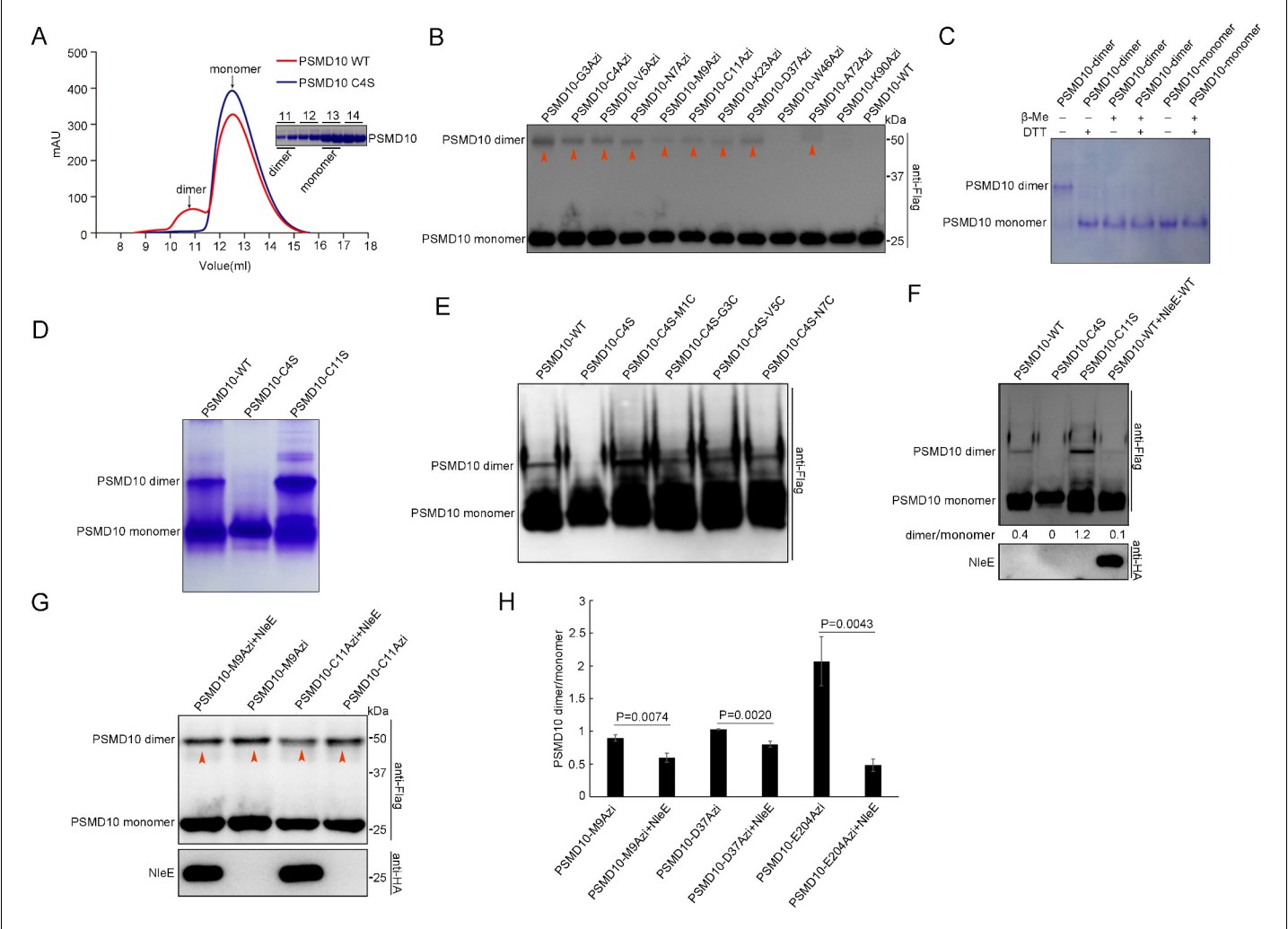

**Figure 5.** NleE suppresses homodimerization of PSMD10. (**A**) C4S mutation suppresses homodimerization of purified PSMD10 protein in vitro. Protein from the dimer fraction of WT PSMD10 shows MW corresponding to the PSMD10 monomer on SDS–PAGE analysis. (**B**) Covalent capture of PSMD10 homodimers in living cells. (**C**) Disulfide bonds stabilize the PSMD10 homodimer in vitro. The dimer and monomer PSMD10 were analyzed on native gel in the absence and presence of 100 mM DTT or 1% β-Me. (**D**) The disulfide bond of Cys4 stabilizes the PSMD10 homodimer in vitro. (**E**) Disulfide bonds at the N-terminus stabilize homodimer of PSMD10. (**F**) NleE suppresses PSMD10 homodimerization in living cells. (**G–H**) NleE partially suppresses PSMD10 homodimer crosslinking by Uaas in living cells. Crosslinking efficiency were quantified (**H**). Quantification data represent the mean ± SD of independent experiments.

The online version of this article includes the following source data and figure supplement(s) for figure 5:

**Source data 1.** Numerical data for *Figure 5H*.
**Source data 2.** Original western blot files for *Figure 5*.
**Figure supplement 1.** Disulfide bonds stabilize the PSMD10 homodimer.
**Figure supplement 1—source data 1.** Original western blot files for *Figure 5—figure supplement 1*.

Cys4 is involved in stabilization of the PSMD10 homodimer, but not in dimerization of PSMD10. Introduction of a disulfide bond at the N-terminus via mutation of M1/G3/V5/N7 to Cys stabilized the Flag-PSMD10 C4S homodimer (*Figure 5E*). These results together demonstrate that the disulfide bond at the N-terminus of PSMD10 plays a vital role in stabilization of the PSMD10 homodimer.

NleE affects the N-terminus of PSMD10, which is involved in stabilization of the PSMD10 homodimer. Therefore, we determined whether NleE mediated suppression of PSMD10 homodimerization. Native gel electrophoresis showed that NleE suppressed PSMD10 homodimerization in cells (*Figure 5F*). The PSMD10 homodimers that captured by Azi were partially impaired by NleE

(*Figure 5G–H*). The results together indicate that NleE affecting on N-terminus of PSMD10 attenuates PSMD10 homodimerization.

## NleE attenuates autophagy by inhibiting the interaction of PSMD10 homodimer with ATG7

PSMD10 has been reported to promote starvation-induced autophagy by interacting with ATG7 to elevate LC3-II formation and by interacting with HSF1 to activate ATG7 transcription (*Luo et al., 2016*). We next examined the binding activity of PSMD10 with ATG7 and HSF1 in the presence of NleE. Immunoprecipitation assays demonstrated that NleE dramatically suppressed the interaction of PSMD10 with ATG7 in living cells (*Figure 6A*), but not the interaction of PSMD10 with HSF1 (*Figure 6B*; *Dawson et al., 2002*; *Luo et al., 2016*; *Wang and Cheng, 2017*). As expected, the mRNA and protein levels of ATG7 were not affected by NleE (*Figure 6C,D*). ATG7 acts as an E1-like enzyme to catalyze the conjugation of LC3 with lipid phosphatidylethanolamine (PE). The colocalization of ATG7 with LC3 puncta was partially suppressed by NleE (*Figure 6E*). Importantly, PSMD10 interaction with ATG7 was suppressed by EPEC in a NleE-dependent manner (*Figure 6F*). Thus, NleE attenuates host autophagy by blocking the interaction of PSMD10 with ATG7.

In addition, we demonstrated that PSMD10 also coimmunoprecipitates with ATG10 and ATG12 in living cells (*Figure 7—figure supplement 1A*). Interestingly, deleting the first ankyrin repeat prevented PSMD10 interaction with ATG7 and ATG10 (*Figure 7A*), but not its interaction with ATG12

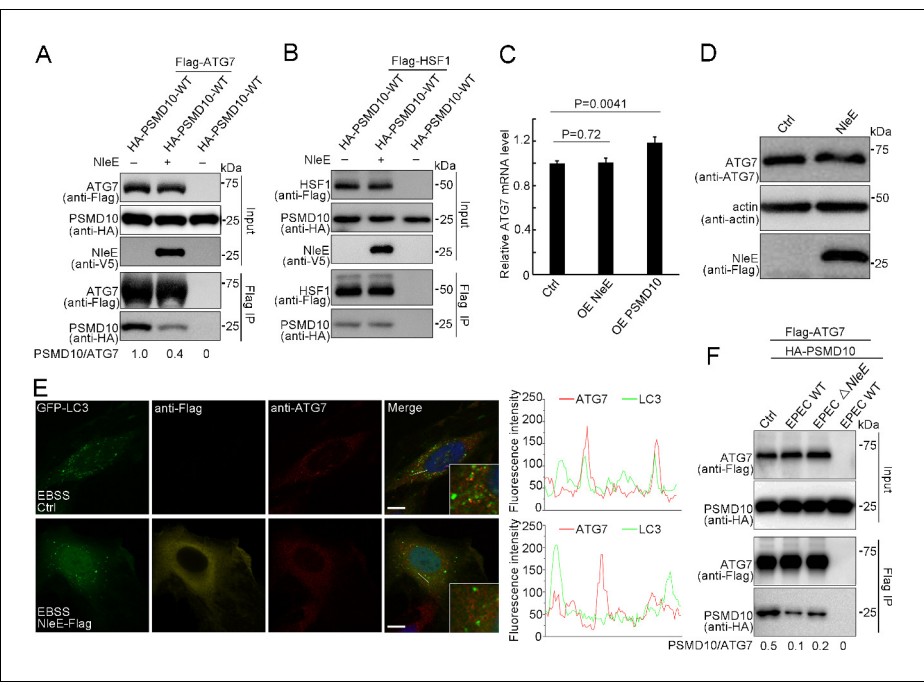

**Figure 6.** NleE suppresses PSMD10 interaction with ATG7. (**A,B**) NleE suppresses the PSMD10 interaction with ATG7, but not the interaction with HSF1. Immunoblots of anti-Flag immunoprecipitates (Flag IP) and total cell lysates (Input) are shown. (**C**) Transcription of ATG7 is not affected by NleE in HeLa cells. Real-time PCR was performed to detect mRNA level of ATG7. Data represent the mean ± SD of three independent experiments. Each sample performs three times technical repeats. (**D**) NleE does not affect the protein level of ATG7 in HeLa cells. Cells were treated with 10 μg/ml LPS for 12 hr and analyzed by immunoblotting using anti-Flag and anti-ATG7 antibodies. (**E**) NleE suppresses ATG7 colocalization with LC3. Representative images are shown. Scale bars, 10 μm. (**F**) PSMD10 interaction with ATG7 was suppressed by EPEC in a NleE dependent manner. HEK293T cells were transiently transfected with PSMD10 or PSMD10 and ATG7. 24 hr later, the cells were infected with EPEC or EPEC ΔNleE for 4 hr. Shown are immunoblots of Flag IP and total cell lysates (Input).

The online version of this article includes the following source data for figure 6:

**Source data 1.** Numerical data for *Figure 6C*.
**Source data 2.** Original western blot files for *Figure 6*.

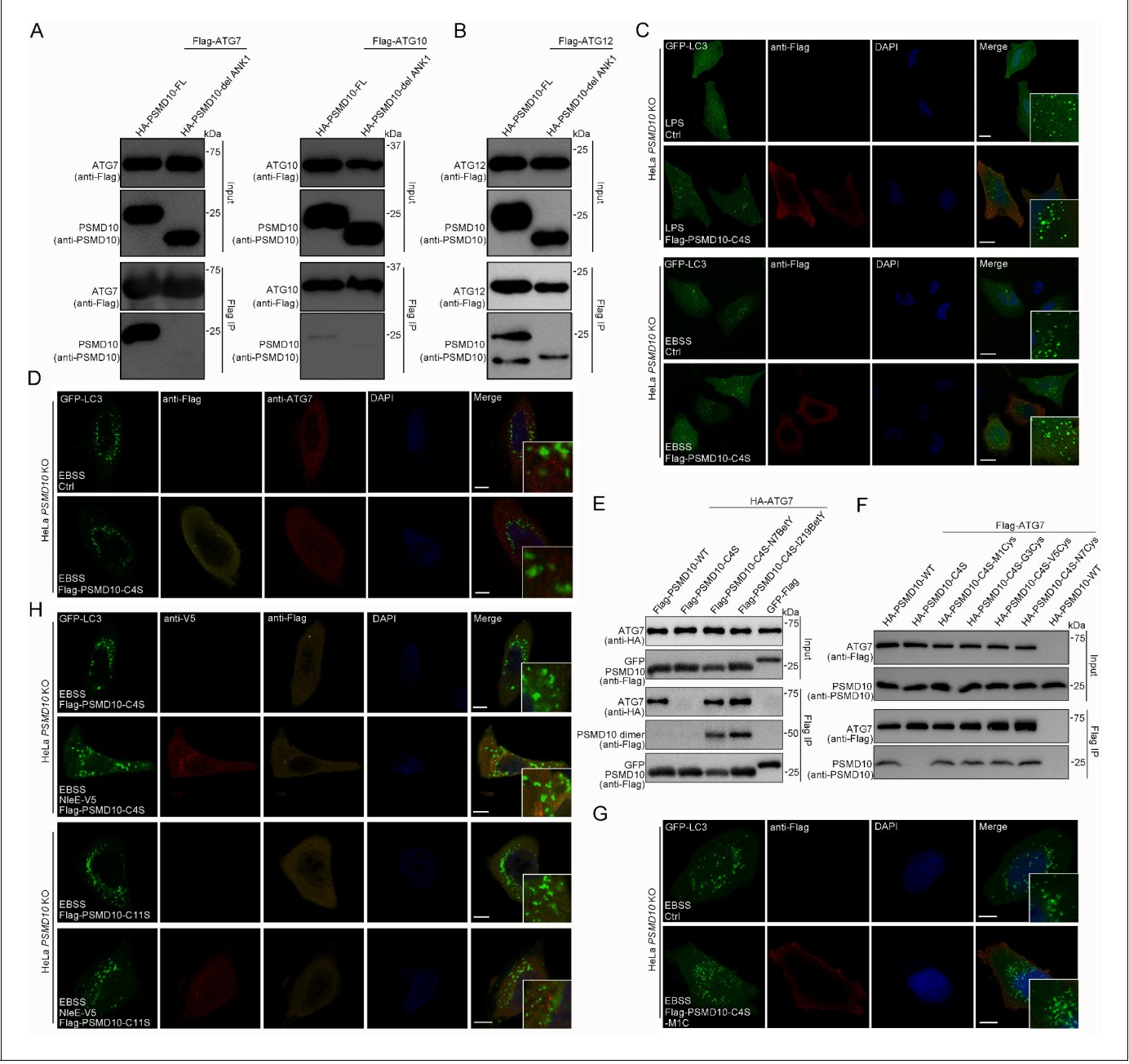

**Figure 7.** Stabilized PSMD10 homodimer plays vital roles in enhancement of autophagy. (**A**) The N-terminus of PSMD10 is indispensable for interacting with ATG7 and ATG10. Shown are immunoblots of anti-Flag immunoprecipitates (Flag IP) and total cell lysates (Input). (**B**) Deletion of the PSMD10 N-terminus does not affect its interaction with ATG12. (**C**) The PSMD10 C4S mutant failed to enhance LPS and starve-induced autophagy in *PSMD10 KO* cells. Scale bars, 10 μm. (**D**) ATG7 and LC3 were not colocalized in *PSMD10 KO* cells expressing the PSMD10 C4S mutant. Representative images are shown. Scale bars, 10 μm. (**E, F**) Stabilized PSMD10 C4S homodimers restore its interaction with ATG7 in living cells. Shown are immunoblots of anti-Flag immunoprecipitates (Flag IP) and total cell lysates (Input). PSMD10 C4S homodimers stabilized by chemical crosslinking (**E**) and disulfide bonds (**F**). (**G**) The stabilized PSMD10 C4S homodimer functions in enhancing autophagy in *PSMD10 KO* cells. Scale bars, 10 μm. (**H**) NleE attenuates host autophagy in a PSMD10 homodimer-dependent manner. The PSMD10 C11S mutant, but not the PSMD10 C4S mutant, rescued the NleE functions in regulating host autophagy in *PSMD10 KO* cells. Scale bars, 10 μm.

The online version of this article includes the following source data and figure supplement(s) for figure 7:

**Source data 1.** Original western blot files for *Figure 7*.

**Figure supplement 1.** Stabilized PSMD10 homodimer is indispensable for interaction with ATG7 and ATG10.

**Figure supplement 1—source data 1.** Original western blot files for *Figure 7—figure supplement 1*.

**Figure supplement 2.** The stabilized PSMD10 homodimer plays vital roles in enhancement of autophagy.

*Figure 7 continued on next page*

*Figure 7 continued*

**Figure supplement 2—source data 1.** Numerical data for *Figure 7—figure supplement 2A*.
**Figure supplement 2—source data 2.** Numerical data for *Figure 7—figure supplement 2B*.
**Figure supplement 2—source data 3.** Numerical data for *Figure 7—figure supplement 2C*.
**Figure supplement 2—source data 4.** Numerical data for *Figure 7—figure supplement 2D*.
**Figure supplement 2—source data 5.** Numerical data for *Figure 7—figure supplement 2E*.
**Figure supplement 3.** PSMD10 homodimer is indispensable for ATG7 dimerization and its interaction with ATG3 and LC3.
**Figure supplement 3—source data 1.** Original western blot files for *Figure 7—figure supplement 3*.

in living cells (*Figure 7B*). Photocrosslinking demonstrated that the last three ankyrin repeats of PSMD10 directly interact with ATG7 (*Figure 7—figure supplement 1B,C*). Three to five ankyrin repeats of PSMD10 were found to be responsible for binding with ATG12 (*Figure 7—figure supplement 1D*). Consistently, PSMD10 C4S mutant failed to interact with ATG7 and ATG10 (*Figure 7—figure supplement 1E,F*), while interacted with Rpt3 (*Figure 3—figure supplement 1D*). Thus, PSMD10 homodimerization may play important roles in autophagy, but not in proteasome. As expected, PSMD10 C4S failed to enhance starvation and LPS-induced autophagy in *PSMD10 KO* cells (*Figure 7C*, *Figure 7—figure supplement 2A,B*). Moreover, PSMD10 C4S also failed to rescue the colocalization defect of ATG7 and LC3 in *PSMD10 KO* cells (*Figure 7D*). Nevertheless, Uaa-mediated chemical crosslinking stabilized the PSMD10 C4S homodimer and restored its binding activity with ATG7 (*Figure 7E*). Disulfide bonds in PSMD10 C4S mutant restored its binding activity with ATG7 (*Figure 7F*) and enhanced autophagy in *PSMD10 KO* HeLa cells (*Figure 7G*, *Figure 7—figure supplement 2C*). Furthermore, PSMD10 C11S, but not PSMD10 C4S, rescued the NleE-mediated autophagy suppression phenotype in *PSMD10 KO* cells (*Figure 7H*, *Figure 7—figure supplement 2D–E*). Altogether, our data indicate that stabilization of the PSMD10 homodimer is essential for its interaction with ATG7 and autophagy promotion. Therefore, NleE-mediated monomeric PSMD10 fails to interact with ATG proteins and promote autophagy.

To determine which region of ATG7 is required for its interaction with PSMD10, a series of ATG7 deletion mutants containing different domains was used (*Figure 7—figure supplement 3A*). The adenylation domain (AD) that mediates ATG7 homodimerization (*Noda et al., 2011*) was found to be responsible for the binding of ATG7 with PSMD10 (*Figure 7—figure supplement 3B*). Notably, homodimerization of ATG7 is thought to play an essential role in LC3 modification (*Noda et al., 2011*). LC3 is initially recognized by the C-terminal tail of ATG7 and transferred to AD (*Noda et al., 2011*). LC3 is then transferred to Atg3, which binds to the opposite protomer of the ATG7 homodimer with much higher efficiency (*Noda et al., 2011*). However, the dimerization of ATG7 was not affected by WT PSMD10, PSMD10 C4S mutant, or NleE (*Figure 7—figure supplement 3C*). The ATG7/LC3 and ATG7/ATG3 interactions were not affected by either WT PSMD10 or PSMD10 C4S mutant (*Figure 7—figure supplement 3D,E*). Therefore, the PSMD10 homodimer enhances autophagy through an unidentified mechanism, rather than through the effects of ATG7 dimerization or ATG7 interaction with LC3 and ATG3.

## Discussion

Autophagy plays a pivotal role in host defense against pathogen infections. The interaction between bacterial pathogens and host autophagy is a mutual process, and many bacteria have developed diverse mechanisms to combat host autophagy. The mechanism of how pathogens affect autophagy in host cells remains largely elusive. In this study, we found that the EPEC effector NleE attenuates autophagosome formation by interacting with host PSMD10. PSMD10 homodimers interact with ATG7 and promote autophagy. The NleE-mediated PSMD10 in a monomeric state attenuates autophagy.

Generally, interactions between host proteins and effectors are weak, dynamic, and transient in living cells. We applied genetically encoded photocrosslinking Uaas to covalently capture the transient interaction partners of NleE in situ. Furthermore, covalent complexes are tolerant to stringent processing, which makes purification under denaturing conditions possible. The dynamic interaction of NleE with the N-terminus of PSMD10 in living cells was revealed by pairwise chemical crosslinking.

Genetical encoding technology is advantageous for capturing dynamic protein–protein interactions in different conformations. Our research represents that genetic encoding technology is a powerful approach to identify unknown interactions and reveal interaction models in mammalian cells. It will convince people to use genetically incorporated photoreactive Uaas, together with pairwise chemical crosslinking in future studies.

Previously, NleE was identified as an SAM-dependent methyltransferase in EPEC. It methylated specific zinc-coordinating cysteines in TAB2/3 and ZRANB3 to disrupt their ubiquitin-chain binding ability. Here, we demonstrated that NleE attenuates autophagy in host cells at the autophagosome formation step in a PSMD10 dependent manner. PSMD10 interacts with ATG7, ATG12, Rpt3, and HSF1. Stabilization of PSMD10 homodimer is required for its interaction with ATG7, but not with ATG12, Rpt3, and HSF1. Homodimerization of ATG7 plays an important role in LC3 modification. The AD of ATG7 that mediates ATG7 homodimerization is also responsible for binding of ATG7 with PSMD10. These observations may explain why NleE-mediated PSMD10 in the monomeric state specifically inhibits the interaction of PSMD10 with ATG7, but not the interaction of PSMD10 with other interaction partners. Our study showed the mechanism by which NleE specifically functions to inhibit autophagy. While inhibition of autophagy may affect host responses toward EPEC. NleE-mediated inhibition of autophagy impairs cytokine IL-6 production in macrophages, which was reported to contribute to host defense through stimulation of acute phase responses, hematopoiesis, and immune reactions.

The mechanism by which the interaction of PSMD10 homodimer with ATG7 enhances autophagy remains to be further investigated. Homodimerization of ATG7 is thought to play an important role in LC3 modification (*Noda et al., 2011*). NleE, *PSMD10 KO* and monomeric C4S mutation did not affect the dimerization of ATG7, excluding the possibility that the PSMD10 homodimer facilitates ATG7 dimerization. Moreover, we did not find that the interactions of ATG7 with LC3 and ATG3 were affected by either WT PSMD10 or the PSMD10 C4S mutant.

Collectively, our research demonstrated that EPEC has evolved strategies to block the host autophagic response partially via PSMD10. Our findings provide new insight in the fight against EPEC infection and provide new potential targets for therapeutic intervention for EPEC and related pathogens. Importantly, NleE is highly conserved across attaching and effacing (A/E) pathogens and has homologs in *Shigella*, raising the possibility that NleE homologs are involved in blocking host autophagic responses.

## Materials and methods

**Key resources table**

| Reagent type (species) or resource | Designation | Source or reference | Identifiers | Additional information |
|---|---|---|---|---|
| Gene (*Homo sapiens*) | ATG7 | National Center for Biotechnology Information | Gene ID: 10533 | |
| Gene (*Homo sapiens*) | PSMD10 | National Center for Biotechnology Information | Gene ID: 5716 | |
| Gene (*Homo sapiens*) | ATG3 | National Center for Biotechnology Information | Gene ID: 64422 | |
| Gene (*Homo sapiens*) | LC3B | National Center for Biotechnology Information | Gene ID: 81631 | |
| Gene (*Homo sapiens*) | ATG10 | National Center for Biotechnology Information | Gene ID: 83734 | |
| Gene (*Homo sapiens*) | ATG12 | National Center for Biotechnology Information | Gene ID: 9140 | |

*Continued on next page*

*Continued*

| Reagent type (species) or resource | Designation | Source or reference | Identifiers | Additional information |
|---|---|---|---|---|
| Gene (*Homo sapiens*) | Rpt3 | National Center for Biotechnology Information | Gene ID: 5704 | |
| Gene (*Homo sapiens*) | TAB2 | National Center for Biotechnology Information | Gene ID: 23118 | |
| Gene (*Homo sapiens*) | HSF1 | National Center for Biotechnology Information | Gene ID: 3297 | Additional information |
| Gene (*Escherichia coli*) | NleE | National Center for Biotechnology Information | Gene ID: NC_011601.1 | |
| Strain, strain background (*Escherichia coli*) | EPEC E2348/69 | National Institute of Biological Sciences | | |
| Strain, strain background (*Escherichia coli*) | DH5α | Tsingke Biotechnology | Catalog # TSV-A07 | Chemically Competent Cell |
| Strain, strain background (*Escherichia coli*) | BL21 | Tsingke Biotechnology | Catalog # TSV-A09 | Chemically Competent Cell |
| Cell line (*Homo sapiens*) | HeLa | National Collection of Authenticated Cell Cultures | Catalog # TCHu187 | |
| Cell line (*Homo sapiens*) | THP-1 | National Collection of Authenticated Cell Cultures | Catalog # TCHu57 | |
| Cell line (*Homo sapiens*) | HEK293T | National Collection of Authenticated Cell Cultures | Catalog # GNHu17 | |
| Antibody | DYKDDDDK-Tag (3B9) Mouse monoclonal antibody | Abmart | RRID:AB_2713960; Catalog # M20008 | Western blot (1:2000); Immunofluorescence (1:500) |
| Antibody | LC3A/B (D3U4C) XP Rabbit monoclonal antibody | Cell Signaling Technology | RRID:AB_2617131; Catalog # 12741S | Western blot (1:2000); Immunofluorescence (1:500) |
| Antibody | ACTB Rabbit monoclonal antibody | ABclonal | RRID:AB_2768234; Catalog # AC026 | Western blot (1:5000) |
| Antibody | V5 tag Rabbit monoclonal antibody | Abmart | RRID:AB_2864358; Catalog # T40006 | Western blot (1:2000); Immunofluorescence (1:500) |
| Antibody | HA-Tag(26D11) Mouse monoclonal antibody | Abmart | RRID:AB_2864345; Catalog # M20003 | Western blot (1:2000); Immunofluorescence (1:500) |
| Antibody | Anti-HA tag Rabbit monoclonal antibody | Abcam | RRID:AB_2864361; Catalog # ab236632 | Western blot (1:2000); Immunofluorescence (1:500) |
| Antibody | HRP Anti-DDDDK tag Goat polyclonal antibody | Abcam | RRID:AB_299061; Catalog # ab1238 | Western blot (1:2000) |

*Continued on next page*

Continued

| Reagent type (species) or resource | Designation | Source or reference | Identifiers | Additional information |
|---|---|---|---|---|
| Antibody | Goat anti-rabbit IgG (H+L), HRP conjugate polyclonal antibody | Proteintech | RRID:AB_2722564; Catalog # SA00001-2 | Western blot (1:2000) |
| Antibody | Anti-ATG7 Rabbit monoclonal antibody | Abcam | RRID:AB_867756; Catalog # ab52472 | Western blot (1:2000); Immunofluorescence (1:500) |
| Antibody | Anti-Gankyrin Rabbit monoclonal antibody | Abcam | RRID:AB_2864359; Catalog # ab188315 | Western blot (1:2000) |
| Antibody | Ub (P4D1) Mouse monoclonal antibody | Santa Cruz Biotechnology | RRID:AB_628423; Catalog # sc-8017 | Western blot (1:2000) |
| Antibody | gankyrin (A-8) Mouse monoclonal antibody | Santa Cruz Biotechnology | RRID:AB_2172940; Catalog # sc-166376 | Western blot (1:2000) |
| Antibody | GST-Tag(12G8) Mouse monoclonal antibody | Abmart | RRID:AB_2864360; Catalog # M20007 | Western blot (1:2000) |
| Recombinant DNA reagent | pCMV and derivatives (plasmids) | This paper | Described in Materials and methods section | |
| Recombinant DNA reagent | pcDNA3.1(+) and derivatives (plasmids) | This paper | Described in Materials and methods section | |
| Recombinant DNA reagent | pLKO.1 and derivatives (plasmids) | This paper | Described in Materials and methods section | |
| Recombinant DNA reagent | px330 and derivatives (plasmids) | This paper | Described in Materials and methods section | |
| Recombinant DNA reagent | pCVD442 and derivatives (plasmids) | This paper | Described in Materials and methods section | |
| Recombinant DNA reagent | pET-28a(+) and derivatives (plasmids) | This paper | Described in Materials and methods section | |
| Recombinant DNA reagent | pGEX-6P-1 and derivatives (plasmids) | This paper | Described in Materials and methods section | |
| Commercial assay or kit | Cell Total RNA Isolation Kit | FOREGENE | Catalog # RE-03113 | |
| Commercial assay or kit | HiScript II Q RT SuperMix (+gDNA wiper) | Vazyme | Catalog # R223-01 | |
| Commercial assay or kit | AceQ Universal SYBR qPCR Master Mix | Vazyme | Catalog # Q511-02 | |
| Commercial assay or kit | Human IL-6 ELISA Kit | ABclonal | Catalog # RK00004 | |
| Commercial assay or kit | Apoptosis and Necrosis Assay Kit | Beyotime Biotechnology | Catalog # C1056 | |
| Chemical compound, drug | LPS | Sigma-Aldrich | Catalog # L3024 | 10 ug/ml for HeLa cells |

Continued

| Reagent type (species) or resource | Designation | Source or reference | Identifiers | Additional information |
|---|---|---|---|---|
| Chemical compound, drug | Phorbol 12-myristate 13-acetate(PMA) | MedChemExpress | Catalog # HY-18739 | 200 ng/ml for THP-1 cells |
| Chemical compound, drug | Chloroquine (CQ) | MedChemExpress | Catalog # HY-17589 | 50 uM for HeLa cells |
| Chemical compound, drug | Azi | SustGreen Tech | CAS # 33173-53-4 | 1 mM for HEK293T cells |
| Chemical compound, drug | DiZPK | SustGreen Tech | Catalog # HBC-066 | 1 mM for HEK293T cells |
| Chemical compound, drug | BetY | SustGreen Tech | CAS # 481052-60-2 | 0.5 mM for HEK293T cells |
| Chemical compound, drug | MG132 | MedChemExpress | Catalog # HY-13259 | 10 uM for HEK293T cells |
| Chemical compound, drug | SAM | Sangon Biotech | Catalog # A506555-0005 | 0.8 mM for in vitro methylation assay |
| Chemical compound, drug | DSS | Sangon Biotech | Catalog # C100015-0100 | 1 mM for extracellular protein crosslinking |
| Chemical compound, drug | DSP | Sangon Biotech | Catalog # C110213-0100 | 1 mM for extracellular protein crosslinking |
| Chemical compound, drug | Wortmannin | selleck | Catalog # S2758 | 1 uM for THP-1 cell |
| Software, algorithm | GraphPad Prism 5 | GraphPad Prism | RRID:SCR_002798; http://www.graphpad.com/ | |
| Software, algorithm | ChemDraw | ChemDraw | RRID:SCR_016768; http://www.perkinelmer.co.uk/category/chemdraw | |
| Software, algorithm | Origin 9 | Originlab | https://www.originlab.com/ | |
| Software, algorithm | ImageJ | ImageJ | RRID:SCR_003070; https://imagej.net/ | |
| Software, algorithm | PyMOL | PyMOL | RRID:SCR_000305; http://www.pymol.org/ | |
| Other | Anti-DYKDDDDK-Tag Mouse Antibody (Agarose Conjugated) | Abmart | Catalog # M20018L | |
| Other | Mounting Medium With DAPI Aqueous, Fluoroshield | Abcam | Catalog # ab104139 | |
| Other | Fluoresbrite BB Carboxylate Microspheres 1.75 μm | Polysciences | Catalog # 17686–5 | |

## Plasmids and antibodies

The mammalian expression vectors for NleE and pHY-XYRS were synthesized by GENERAL BIOSYS-TEMS. The pHY-XYRS plasmid contains three copies of the suppressor tRNA driven by the human U6 promoter and one copy of XYRS driven by a PGK promoter (*Wang et al., 2007*). pIRE4-Azi was purchased from addgene. cDNAs for LC3, PSMD10, ATG7, HSF1, ATG3, TAB2 NZF domain (residues 665–693) and Rpt3 were amplified from HEK293T cDNA library. Genes were inserted into pET-28a (+) or pGEX-6P-1 for bacterial expression and inserted into pcDNA3.1(+) or pCMV-HA/Flag for mammalian expression. pLKO.1-CMV, pMD2.G, and psPAX2 plasmids were used for lentiviral production. PX330 and PUC19 plasmids were used to construct knockout cells. All point mutations were generated using the Mut Express MultiS Fast Mutagenesis Kit V2. All plasmids were assembled by standard cloning methods and confirmed by DNA sequencing.

Antibodies against PSMD10(sc-166376) and ubiquitin(sc-8017) were purchased from Santa Cruz. Antibodies against Flag-tag(M20008), HA-tag(M20003), V5-tag(T40006), and GST-tag(M20007) were purchased from Abmart. Antibodies against ATG7(ab52472), PSMD10(ab188315), Flag-tag (HRP) (ab1238), and goat anti-mouse IgG H and L (HRP) (ab6789) were purchased from Abcam. Anti-LC3A/B (4108) antibody was purchased from CST. Anti-ACTB(AC026) antibody was purchased from ABclonal. Anti-HA (D110004) antibody was from Sangon Biotech. HRP-conjugated goat anti-rabbit IgG(H+L) (SA00001-2) was purchased from Proteintech. All secondary antibodies for immunofluorescence staining were purchased from Thermo Fisher Scientific.

## Cell culture, transfection, and immunoblotting

HEK293T, THP-1, and HeLa cells were originally obtained from National Collection of Authenticated Cell Cultures. THP-1 cells were cultured in RPMI 1640 medium supplemented with 10% fetal bovine serum (FBS) and maintained at 37°C under humidified conditions with 5% $CO_2$. Other cells were cultured in Dulbecco's modified Eagle's medium (DMEM) supplemented with 10% fetal bovine serum (FBS) and maintained at 37°C under humidified conditions with 5% $CO_2$. Transient transfection was performed using Lipofectamine 2000 following the manufacturer's instructions. All cell lines tested negative for mycoplasma contamination regularly based on PCR assays.

For immunoblotting analysis, cells were lysed in lysis buffer (25 mM Tris–HCl pH 7.4, 150 mM NaCl, 0.5% Triton X-100, 5% glycerin) supplemented with 1% protease inhibitor cocktail. Samples were separated on 10% SDS–PAGE gels and transferred onto PVDF membranes. Membranes were blocked with 5% milk in TBST, incubated with primary antibody, and then incubated with HRP-conjugated secondary antibodies. Proteins were visualized using Clarity Western ECL Substrate on a Clinx ChemiScope 5300 Imaging System.

## EPEC knockout, culture, and infection

EPEC *E. coli* E2348/69 was used as the wild-type strain. NleE deletion was constructed by homologous recombination using the positive-selection suicide vector pCVD442. The 600 bp 5′ and 600 bp 3′ flanking sequences of the NleE open-reading frame were cloned into the pCVD442 plasmid. The vector was introduced into SM10(λπ) and transferred into EPEC E2348/69 by filter mating, with selection for nalidixic acid and ampicillin resistance. The positive colonies were further counterselected on LB plates (without NaCl) supplemented with 20% sucrose. Sucrose-resistant colonies were picked and tested for the loss of the intact NleE gene by PCR and DNA sequencing.

Infection was performed with macrophage. THP-1 cells were treated with 200 ng/ml phorbol 12-myristate 13-acetate (PMA) for 48 hr to induce macrophages. Bacteria were grown overnight in LB medium with shaking at 37 °C at 220 rpm and then subcultured (1:30) in fresh RPMI 1640 medium for 2–3 hr. Bacterial culture was added to macrophage cells at an MOI = 100 and incubated for 3 hr. Cells were fixed in 4% paraformaldehyde and immunostained with anti-LC3 antibody.

## Immunofluorescence

Cells were fixed in 4% paraformaldehyde for 30 min and permeabilized with 0.5% Triton X-100 for 30 min at room temperature. Samples blocked with 10% normal goat serum in PBS were incubated with the indicated primary antibodies overnight at 4°C. Corresponding secondary antibodies were incubated for 1 hr at room temperature. Coverslips were mounted onto microscope slides with

mounting medium containing DAPI. Images were acquired using a confocal laser scanning microscope (ZEISS LSM 880).

## Autophagy induction

Cells were treated with EBSS for 6 hr or 10 µg/ml LPS for 12 hr for autophagy induction. Polystyrene beads transfection was performed to induce xenophagy (*Xu et al., 2019*). Fluoresbrite BB carboxylate microspheres (1.75 µm) were washed three times with PBS and resuspended in PBS. Approximately $8.5 \times 10^6$ beads were transfected into cells in 12-well plates using Lipofectamine 2000. Four hours after transfection, the cells were washed three times with PBS and then fixed in 4% paraformaldehyde for further analysis.

## Immunoprecipitation and pulldown assays

For anti-Flag immunoprecipitation, cells were collected 48 hr after transfection and lysed in lysis buffer (25 mM Tris–HCl pH 7.4, 150 mM NaCl, 0.5% Triton X-100, 5% glycerin) supplemented with 1% protease inhibitor cocktail for 30 min on ice. Anti-FLAG affinity beads were added to the supernatant and incubated for 6 hr at 4°C with constant rotation. The anti-FLAG affinity b hed four times with lysis buffer and then denatured by SDS loading buffer for subsequent immunoblotting.

For MS identification of NleE crosslinked proteins, four 10 cm dishes of HEK293T cells were cotransfected with pIRE4-Azi and pcDNA3.1-NleE-K219TAG-Flag-6xHis or pcDNA3.1-NleE-WT-Flag-6xHis. Cells were cultured in the presence of 1 mM Azi for 48 hr and irradiated with a UVP crosslinker for 20 min on ice before harvesting. Cells were lysed in lysis buffer (50 mM Tris–HCl pH 7.4, 150 mM NaCl, 0.5% Triton X-100) supplemented with protease inhibitor cocktail for 30 min on ice. The supernatant was incubated with anti-FLAG affinity beads for 6 hr at 4°C with constant rotation. The immunoprecipitated proteins were eluted with buffer D (50 mM Tris–HCl pH 7.4, 150 mM NaCl, 8 M urea, 5 mM β-Me, 10 mM Imidazole). Ni-NTA resin was added to the elution and incubated for 2 hr at 4°C with constant rotation. The Ni-NTA resin was washed four times with lysis buffer D and two times with lysis buffer E (50 mM Tris–HCl pH 7.4, 150 mM NaCl, 5 mM β-Me, 10 mM Imidazole). The proteins were eluted with elution buffer (50 mM Tris–HCl pH 7.4, 150 mM NaCl, 300 mM Imidazole) for subsequent MS analysis.

For pulldown assays, the GST-tagged proteins were immobilized with glutathione resin in binding buffer (50 mM Tris–HCl pH 7.4, 150 mM NaCl, 0.5% NP-40) and then incubated with binding proteins for 4 hr at 4°C. Beads were washed four times with binding buffer and denatured by SDS loading buffer. Samples were analyzed using immunoblotting.

## Stable cell line construction

To generate stably expressed cells, the pLKO.1-CMV vector together with pSPAX2 and pMD2.G vectors was transfected into HEK293T cells. The supernatants containing lentiviral particles were collected 24 hr and 48 hr after transfection and filtered through a 0.45 µm filter. Then, lentiviral particles were added to the infected cells (70% confluence) in six-well plates. Cells were incubated overnight at 37°C in a 5% $CO_2$ incubator. The solution was changed to fresh media, and the cells were cultured for another 24 hr. Then, 2 µg/ml puromycin was added for selection. One week later, the cells were lifted and tested for expression of the transgene.

## CRISPR/Cas9 knockout cell lines

Construction of knockout cells by CRISPR/Cas9 was performed as described (*Cong et al., 2013*). The guide RNA (gRNA) target sequences used for *PSMD10* were TATTCTGGCCGATAAATCCC and CTTCATATTGCGGCTTCTGC. The PX330 and HR template vectors were cotransfected into cells. Forty-eight hours after transfection, cells were subcultured into complete DMEM with 2 µg/ml puromycin. Two weeks later, the cells were lifted, diluted, and seeded in 96-well plates. Single-cell clones were sequenced, and the expression of endogenous PSMD10 was tested by immunoblotting analysis.

## In vitro methylation assay

In vitro methylation assay was performed as described (*Zhang et al., 2012*). 4 µg of NleE was incubated with 2 µg of GST-PSMD10 or GST-TAB2-NZF for 30 min at 37°C in 20 µl methylation buffer

(50 mM Tris–HCl pH 7.4, 150 mM NaCl, 5 mM DTT, 0.1% NP-40, and 0.8 mM S-adenosylmethionine). The reaction mixtures were separated on a 15% Native–PAGE gels or 12% SDS–PAGE gels, followed by Coomassie blue staining and MS analysis.

## Uaa incorporation and crosslinking

pIRE4-Azi, pCMV-MbPylRS(DiZPK) or pHY-XYRS plasmids were cotransfected with the pcDNA3.1-NleE-XTAG-Flag vector into HEK293T cells. The final concentration of Uaas in culture medium was 1 mM for Azi, 1 mM for DiZPK, and 0.5 mM for BetY. Forty-eight hours after transfection, cells were harvested and irradiated with a UVP crosslinker for 20 min on ice for Azi and DiZPK. The cells were lysed in lysis buffer C (50 mM Tris–HCl pH 7.4, 150 mM NaCl, 0.5% Triton X-100) and denatured by SDS loading buffer. Samples were then analyzed using anti-FLAG immunoblotting.

## Quantitative real-time PCR

Total RNA was isolated using a Cell Total RNA Isolation Kit according to the manufacturer's protocol. Total RNA was subjected to reverse transcription into cDNA by using HiScript II Q RT SuperMix (+gDNA wiper). Quantitative real-time PCR was performed using AceQ Universal SYBR qPCR Master Mix on a Bio-Rad CFX96 PCR System. The housekeeping gene, ACTB, was used as controls. The sequences of the primer sets were 5-CTGGCCGGGGATGAGATTGTAAAAG-3 and 5-CGGTGCATTGCTGTAGCCTCATAA-3 for PSMD10; 5-CCCAAGGCCAACCGCGAGAAGATG-3 and 5-GTCCCGGCCAGCCAGGTCCAGA-3 for ACTB; and 5-TGCTATCCTGCCCTCTGTCTT-3 and 5-TGCCTCCTTTC TGGTTCTTTT-3 for ATG7.

## Protein expression and purification

All proteins were expressed in *E. coli* BL21 (DE3) cells harboring expressing plasmids. Protein expression was induced with 0.5 mM IPTG for 16 hr at 18℃. Affinity purification of GST-tagged proteins and His-tagged proteins was performed using glutathione resin and Ni-NTA resin, respectively. Purified proteins were concentrated using Millipore Amicon Ultra.

## Size exclusion chromatography

Ten milligrams of purified PSMD10 protein was loaded onto a SuperdexTM 200 Increase 10/300 GL column that was pre-equilibrated with equilibrium buffer (50 mM Tris–HCl pH 7.4, 150 mM NaCl). Chromatography was performed on an NGC Quest 10 Chromatography System at a flow rate of 0.5 ml/min at 4℃.

## Mass spectrometry

For MS identification of NleE crosslinked proteins, LC–MS/MS analysis was executed using an EASY-NLC 1000 nanoflow LC instrument coupled to a Q Exactive quadrupole-Orbitrap mass spectrometer (Thermo Fisher Scientific). MS spectra were acquired from 350 m/z to 1600 m/z with a resolution of 70,000 at m/z = 200. The automatic gain control (AGC) value was set at 3e6, with a maximum fill time of 20 ms. For MS/MS scans, the top 20 most intense precursors were selected with a 1.6 m/z isolation window and fragmented with a normalized collision energy of 27%. The AGC value for MS/MS was set to a target value of 5e4, with a maximum fill time of 100 ms. Raw files were analyzed against the Swiss-Prot human protein sequence database (20413 entries, 2017/01/14) in MaxQuant (version 1.6) with a reverse decoy database with a false discovery rate (FDR) < 1%. Searches were carried out with a precursor peptide mass tolerance of 10 ppm and a fragment ion mass tolerance of 0.02 Da. Two missed trypsin cleavages were allowed in these searches. Cysteine carbamidomethylation was set as a fixed modification. Oxidation of methionine, acetylation on lysine, and protein N-terminal acetylation were set as variable modifications.

For the in vitro methylation assay, the samples were sent to Bio-Tech Pack Technology for MS analysis. Intact proteins were analyzed by LC–MS using a Thermo Q Exactive mass spectrometer coupled to an ACQUITY UPLC system. Ten micrograms of protein samples were injected by an autosampler and separated on an ACQUITY UPLC Protein BEH C4 Column (300 Å, 1.7 μm, 2.1 mm × 50 mm) by a reverse-phase gradient of 0–80% acetonitrile for 15 min. Mass calibration was performed right before the analysis. MS spectra were acquired from 400 m/z to 5000 m/z. Protein spectra were

averaged, and the charge states were deconvoluted using Protein Deconvolution software. Carbamidomethylation and methylation on cysteine were set as variable modifications.

For the living cell methylation assay, the immunoprecipitated proteins were sent to Shanghai Bioprofile Technology for MS analysis. The samples were digested with trypsin for 16–18 hr at 37°C. The digested peptides were loaded onto an EASY-NLC 1200 UHPLC system. The eluted peptides were sprayed into the Q Exactive Plus LC-MS system. The mass spectrometer was operated in data-dependent mode with one MS scan followed by 20 HCD (high-energy collisional dissociation) MS/MS scans for each cycle. MS spectra were acquired from 300 m/z to 1800 m/z with a resolution of 70,000 at m/z = 200. The AGC value was set at 1e6, with a maximum fill time of 50 ms. For MS/MS scans, the top 20 most intense precursors were selected. The AGC value for MS/MS was set to a target value of 1e5, with maximum fill time of 50 ms. Database searches were performed by MaxQuant 1.6.0.16 against the UniProt *Homo sapiens* Protein Database with a false discovery rate (FDR) < 1%. Searches were carried out with a precursor peptide mass tolerance of 20 ppm. Two missed trypsin cleavages were allowed in these searches. Carbamidomethylation and methylation on cysteine residues were set as variable modifications.

### IL-6 secretion assay

THP-1 cells were treated with 200 ng/ml Phorbol 12-myristate 13-acetate (PMA) for 48 hr. Macrophages were treated with 1 μM Wortmannin for 2 hr to inhibit autophagy. Bacteria were grown overnight in LB medium with shaking at 37°C at 220 rpm and then subcultured (1:30) into fresh RPMI 1640 medium for 2–3 hr. Bacterial culture was added to cells at an MOI = 100 and incubated for 4 hr. Cell culture supernatant was collected andIL-6 secretion was measured by ELISA. Cells were harvested in 1× SDS loading buffer for subsequent immunoblotting.

### Statistical analysis

All experiments were performed in at least three triplicates. Quantitative results are reported as the mean ± SD. Statistical significance between samples was determined by two-way ANOVA followed by multiple comparisons using GraphPad Prism 6.0 software.

## Acknowledgements

We thank Prof. PR Chen (Peking University) for kindly providing the plasmids pCMV-MbPylRS (DiZPK), Prof. F Shao (National Institute of Biological Sciences, Beijing) for kindly providing the Enteropathogenic *E. coli* E2348/69 strain, Prof. YQ ZhU (Zhejiang University) and Prof. N Dong for kindly providing the pCVD442 vector and SM10(λπ) strain, and Prof. H Zhang (Institute of Biophysics, Chinese Academy of Sciences) for discussions and critical reading of the manuscript. This work was supported by grant to HY from the National Key Research and Development Program of China (2018YFC1002802 and 2018YFA0109200) the National Natural Science Foundation of China (31872727), and 1.3.5 project for disciplines of excellence, West China Hospital, Sichuan University (ZYYC20020).

## Additional information

### Funding

| Funder | Grant reference number | Author |
| --- | --- | --- |
| National Key Research and Development Program of China | 2018YFC1002802 | Haiyan Ren |
| National Key Research and Development Program of China | 2018YFA0109200 | Haiyan Ren |
| National Natural Science Foundation of China | 31872727 | Haiyan Ren |
| 1.3.5 project for disciplines of excellence, West China Hospi- | ZYYC20020 | Haiyan Ren |

tal, Sichuan University

The funders had no role in study design, data collection and interpretation, or the decision to submit the work for publication.

### Author contributions
Jingxiang Li, Data curation, Validation, Investigation; Shupan Guo, Fangni Chai, Qi Sun, Pan Li, Xiaoxiao Ouyang, Zhihui Zhou, Li Zhou, Investigation; Li Gao, Data curation, Investigation, performed mass spectrum analysis; Lunzhi Dai, Data curation, performed mass spectrum analysis; Wei Cheng, Shiqian Qi, Kefeng Lu, provided expertise and feedback; Haiyan Ren, Data curation, Funding acquisition, Writing - original draft, Project administration, Writing - review and editing

### Author ORCIDs
Jingxiang Li https://orcid.org/0000-0002-9747-758X
Haiyan Ren https://orcid.org/0000-0001-9995-3255

### Decision letter and Author response
Decision letter https://doi.org/10.7554/eLife.69047.sa1
Author response https://doi.org/10.7554/eLife.69047.sa2

## Additional files

### Supplementary files
- Supplementary file 1. List of primers used in this paper.
- Transparent reporting form

### Data availability
All data generated or analysed during this study are included in the manuscript and supporting files.

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
