## [Decision Letter]

**Acceptance summary:**

This manuscript reveals a new mechanisms used by pathogenic bacteria to escape the defense of the attacked cells. The claims are well supported by a conspicuous bulk of solid biochemical experiments both in vitro and in live cells. The findings will be of large interest in the field of autophagy and in general for studies of host-pathogens interactions.

**Decision letter after peer review:**

Thank you for submitting your article "Genetically incorporated crosslinkers reveal NleE attenuates host autophagy dependent on PSMD10" for consideration by *eLife*. Your article has been reviewed by 3 peer reviewers, including Ivan Dikic as the Reviewing Editor and Reviewer #1, and the evaluation has been overseen by Dominique Soldati-Favre as the Senior Editor.

Summary:

The mechanisms by which bacterial effectors interact with host factors to modulate autophagy are largely unknown. This paper reveals that the Enteropathogenic *Escherichia coli* (EPEC) effector NleE regulates autophagosome formation in host cells by suppressing homodimerization of PSMD10, which is required for its interaction with ATG7 and promotion of autophagy. The claims are well supported by a series of solid biochemical experiments both in vitro and in live cells. The findings will be of interest to researchers in the field of autophagy and in general for scientists interested in host-pathogens interactions.

Essential Revisions:

1. Does NleE affect basel levels of autophagy or specifically only during stimulation? The authors should assess if inhibition of autophagy have any effects on host responses toward EPEC, such as cytokine production and cell death. This should also be discussed in the introduction and the discussion part.

2. The lack of evidence showing the involvement of PSMD10 in NleE mediated autophagy suppression in the context of EPEC infection. Assays of homodimerization of PSMD10 and interaction between PSMD10 and ATG7 should be repeated under the conditions of EPEC infection, preferably with both WT and NleE mutants.

3. Does C4S mutation affect PSMD28 expression pattern? Does the NleE K219 mutation affect its binding to PSMD10?

Please also read remaining comments as there are several examples where to improve figures, quantification and also presentation quality of data.

*Reviewer #1:*

The authors identified the interacting partner of NleE inside the host cell, which is PSMD10. More detailed molecular characterization was then carried out. It was found that NleE suppresses homodimerization of PSMD10 and its binding to ATG7.

One of the major strengths of the study is the use of genetically incorporated crosslinkers for the identification of interacting partners. Traditional immunoprecipitation and manuscript experiments have difficulties identifying weak and transient interaction or low abundance proteins, such as bacterial virulence factors inside the host cells. Covalent capture using genetically incorporated crosslinkers helped to improve the specificity, reliability and sensitivity and indeed with that the authors were able to reveal PSMD as an interacting partner of NleE. This method could potentially be applied to study other bacterial virulence factors.

One interesting implication of the current study is the possibility of NleE's involvement in neurodegenerative disease. It is known that inflammation and gut dysbiosis are correlated with the occurrence of neurodegenerative diseases, such as Alzheimer's disease. Bacteria are able to influence the brain by inducing neuroinflammation or by delivering virulence factor into into the central nervous system via outer membrane vesicles (OMVs). Whether NleE from *E. coli* can suppress the autophagy process outside the gastrointestinal system is an interesting question and merits further investigation.

My major criticism is the lack of evidence showing the involvement of PSMD10 in NleE mediated autophagy suppression in the context of EPEC infection. Experiments concerning regulation of autophagy by NleE and PSMD10 were solely done on overexpression experiments without the actual bacteria. This makes the conclusion less convincing. Also, the authors did not demonstrate and/or explain why autophagy is important for the host defense against *E. coli*, especially because *E. coli* is extracellular. This decreases the impact of the work.

There are also essential questions that are crucial for the completeness of the study but the authors failed to address. First, does NleE affect basel levels of autophagy or specifically only during stimulation? For example, in Figure 1B it seems that NleE expression alone without other stimulation is able to suppress LC3-II but the quantification is missing, making it impossible to be interpretated. Also, is the methyltransferase activity of NleE required for the suppression of autophagy? Based on the data, PSMD10 is not methylated by NleE. However, the mutants of NleE which are unable to methylate also lost their activities to bind PSMD10. One possibility is that NleE methylates another factor which is required for the binding. Lastly, how is the proteosome system affected NleE? Proteosome and autophagy are two closely connected systems. Since PSMD10 is an important component of the proteosome system, it is surprising that NleE does not have any effects on the proteosome.

I believe the study is novel and bring a significant advancement to the field. I would recommend it to be published in *eLife*, if all the following issues are addressed.

1. The authors did not demonstrate and/or explain why autophagy is important for the host defense against *E. coli*.

The authors should assess if inhibition of autophagy have any effects on host responses toward EPEC, such as cytokine production and cell death. This should also be discussed in the introduction and the discussion part.

2. The lack of evidence showing the involvement of PSMD10 in NleE mediated autophagy suppression in the context of EPEC infection.

Assays of homodimerization of PSMD10 and interaction between PSMD10 and ATG7 should be repeated under the conditions of EPEC infection, preferably with both WT and NleE mutants.

3. Does NleE affect basel levels of autophagy or specifically only during stimulation?

Please provide quantification of autophagic activities under unstimulated conditions (e.g. Figure 1B). Also, there is a general omission of quantification of immunofluorescence data, such as in Figure 6 and 7. The missing quantification makes the readers unable to interpretate the data.

4. Is the methyltransferase activity of NleE required for the suppression of autophagy?

Preferably, mutant strains of EPEC should be constructed and the effects on host cell autophagy should be studied. Overexpressing the mutated NleE which loses the methyltransferase activity is also acceptable.

5. How is the proteosome system affected by NleE?

Please provide data showing whether NleE has an effect on the proteosome, e.g. the levels of ubiquitinated proteins.

*Reviewer #2:*

This work unveils a previously unknown mechanism that of Enteropathogenic *E. coli* uses to suppress the autophagy response of the host organism. The mechanism is mediated by the interaction of the methyltransferase NleE of the bacteria with PSMD10 of the host. After demonstrating that NleE blocks autophagosome formation in several cell lines, authors apply genetically incorporated photo- and chemical crosslinkers to identify the target of NleE in the mammalian cell and the topology of the associated complex. They can clearly show that NleE interacts with the N-terminus PSM10 and hamper its physiologic homodimerization, which in turn prevents PSMD10 from interacting with autophagy-related proteins such as ATG7. Thus, the effect of NleE in blocking autophagy is independent from its enzymatic activity, but rely on a previously unknown feature.

The amount of new information about the molecular functioning of the investigated proteins reported here is impressive. The described experiments are well designed and look very solid. Especially the biochemically experiments with genetically encoded crosslinkers are a beautiful application of this method to answer biological question. Many studies about protein-protein interactions rely of IPs and in many cases only in vitro experiments. This work presents direct evidences of interactions from crosslinking events happening in the live cells, which represents a much higher quality.

Overall the work is well organized, the manuscript is clearly structured and the flow of the general story can be followed quite well. However, I recommend to improve some sections to make important details more easily understandable for the reader. There are also some language issues. Specific comments:

It would be helpful for the general reader to explain earlier that NleE is a methyltransferase. I suggest to insert this information already in the introduction. For instance, the sentence that is now at page 6, line 167/168 could be transferred to the end of line 46 of page 1.

Last paragraph of the introduction. The first sentence (page 3 line 70-71) is more or less a repeat of page 1 line 45/46. I would remove this sentence and substitute it with the last sentence of this paragraph.

Page 4 line 110. What is a "synthetase-tRNATyr suppression system"? either author state correctly that they use an orthodona amino acyl-tRNA synthetase(tRNA pair derived from the pair that incorporates Tyrosin in *E. coli*, or they just leave this detail, and may add a couple of words in the method section.

Page 5 line 141. SAM should be described already here, without waiting until page 6 lines 167/168. This is also a reason because NleE should be described before, otherwise the general reader does not understand what is SAM doing here.

I find the sentences from page 6 line 172 "our previous results… "until the end of the paragraph at page 7 line 176 more confusing than helpful. I recommend to remove these comments as they are unnecessary and not very clear.

Page 7 line 183. Before describing results, it should be explained that BetY was incorporated at several sites of NleE and BetY mutants were reacted with PSMD10. It should also be explained whether the Cys residues in PSMD10 are native or were inserted on purpose. Overall the paragraph between lines 180-191 of page 7 contains beautiful results but could be written in a clearer way.

*Reviewer #3:*

Autophagy can selectively capture invading bacteria and deliver them to lysosomes for degradation. Meanwhile, bacteria have evolved various strategies to evade autophagy surveillance. Bacterial virulence effectors can inhibit autophagy via different mechanisms, including blocking the autophagy induction signal, impairing autophagy recognition, attenuating the function of autophagy proteins, or blocking the maturation of bacterium-containing autophagic vacuoles. However, the mechanism by which bacterial effectors interact with host factors to modulate autophagy remains largely unknown. In this paper, the authors first demonstrated that the Enteropathogenic Escherichia. coli (EPEC) effector NleE regulates autophagosome formation in host cells. They then employed genetically incorporated crosslinkers and identified the 26S Proteasome Regulatory Subunit 10 (PSMD10) as a direct interaction partner of NleE in living cells. They further showed that NleE impairs autophagosome formation by suppressing homodimerization of PSMD10, which is required for its interaction with ATG7 and promotion of autophagy. This study reveals a novel mechanism by which a bacterial virulent effector inhibits host autophagy.

---

## [Author Response]

Essential Revisions:1. Does NleE affect basel levels of autophagy or specifically only during stimulation? The authors should assess if inhibition of autophagy have any effects on host responses toward EPEC, such as cytokine production and cell death. This should also be discussed in the introduction and the discussion part.

NleE affected basel-level autophagy (Figure 1D).

We verified that EPEC ΔNleE infection results in higher cytokine production than EPEC infection in macrophages expressing IKKb^CA^ (S177ES181E) mutant (Figure 1—figure supplement 1B). The IKKb^CA^ mutant constitutively active the NF-κB pathway downstream of TAB2/TAB3. Therefore, this result indicated that NleE affects host cytokine production independent of NleE function of suppressing TAB2/TAB3. In addition, wortmannin treatment attenuates IL-6 production during EPEC infection (Figure 1—figure supplement 1C), which indicated that inhibition of autophagy affects host responses to EPEC.

We did not find obvious difference in cell death during EPEC infection and EPEC ΔNleE infection for 4h, regardless presence of autophagy inhibitor or not (Author response image 1). We discussed effect of inhibition of autophagy on host responses toward EPEC in the introduction and the discussion part in the new manuscript.

**Author response image 1. sa2fig1:** Cell death was not affected by NleE.

2. The lack of evidence showing the involvement of PSMD10 in NleE mediated autophagy suppression in the context of EPEC infection. Assays of homodimerization of PSMD10 and interaction between PSMD10 and ATG7 should be repeated under the conditions of EPEC infection, preferably with both WT and NleE mutants.

Our new data showed that EPEC infection partially suppress interaction between PSMD10 and ATG7 (Figure 6F). While interaction between PSMD10 and ATG7 enhanced during EPEC ΔNleE mutant infection (Figure 6F). We also showed that EPEC infection fail to suppress autophagy of the PSMD10 KD macrophages, indicating that PSMD10 was involved in NleE mediated autophagy suppression during EPEC infection (Figure 3K-L).

3. Does C4S mutation affect PSMD28 expression pattern? Does the NleE K219 mutation affect its binding to PSMD10?

C4S mutation does not affect PSMD28 expression pattern (Author response image 2).

Our crosslinking experiments showed that NleE K219 mutation does not affect its binding with PSMD10 (Author response image 2).

**Author response image 2. sa2fig2:** (A) Expression pattern of PSMD28 C4S mutant. (B) K219 mutation does not affect NleE interaction with PSMD10.

Reviewer #1:1. The authors did not demonstrate and/or explain why autophagy is important for the host defense against E. coli.The authors should assess if inhibition of autophagy have any effects on host responses toward EPEC, such as cytokine production and cell death. This should also be discussed in the introduction and the discussion part.

Please see responses in Essential Revisions question one.

2. The lack of evidence showing the involvement of PSMD10 in NleE mediated autophagy suppression in the context of EPEC infection.Assays of homodimerization of PSMD10 and interaction between PSMD10 and ATG7 should be repeated under the conditions of EPEC infection, preferably with both WT and NleE mutants.

Please see responses in Essential Revisions question two.

3. Does NleE affect basel levels of autophagy or specifically only during stimulation?Please provide quantification of autophagic activities under unstimulated conditions (e.g. Figure 1B).

Please see responses in Essential Revisions question one.

Also, there is a general omission of quantification of immunofluorescence data, such as in Figure 6 and 7. The missing quantification makes the readers unable to interpretate the data.

We quantified immunofluorescence data in Figure 6. Quantification data for Figure 7 was shown in Figure 7—figure supplement 2.

4. Is the methyltransferase activity of NleE required for the suppression of autophagy?Preferably, mutant strains of EPEC should be constructed and the effects on host cell autophagy should be studied. Overexpressing the mutated NleE which loses the methyltransferase activity is also acceptable.

Our data showed that expression level of NleE mutants is very low in EPEC ΔNleE and host cells, comparing with WT NleE (Author response image 3). They failed to suppress autophagy (Author response image 3). In addition, all of these three NleE mutants failed to crosslink with PSMD10 (Figure 2—figure supplement 2F-G) which may cause by structure changes of NleE mutants, but not loss of the methyltransferase activity. We failed to detect PSMD10 methylated by NleE (Figure 4—figure supplement 1B-C). Altogether, we tend to believe that methyltransferase activity of NleE may not require for the suppression of autophagy.

**Author response image 3. sa2fig3:** (A) Expression level of NleE mutants in EPEC. (B) Expression level of NleE R107A mutant in Hela cells. (C) NleE R107A mutant fails to suppress host autophagy during infection. (D) Overexpression of NleE mutant fails to suppress autophagy in HeLa cells.

5. How is the proteosome system affected by NleE?Please provide data showing whether NleE has an effect on the proteosome, e.g. the levels of ubiquitinated proteins.

We did not find obvious differences in ubiquitination levels in cells with and without NleE (Author response image 4), suggesting that NleE may not affect the proteosome system. The result is consistent with that PSMD10 interaction with 26S proteasome AAA-ATPase subunit Rpt3 was not affected by NleE (Figure 3—figure supplement 1B-C). Consistently, PSMD10 C4S mutant still interaction with Rpt3 (Figure 3—figure supplement 1D). Thus, NleE-mediated monomeric PSMD10 does not affect the proteasome function.

**Author response image 4. sa2fig4:** Ubiquitination level was not affected by NleE.

Reviewer #2:Specific comments:It would be helpful for the general reader to explain earlier that NleE is a methyltransferase. I suggest to insert this information already in the introduction. For instance, the sentence that is now at page 6, line 167/168 could be transferred to the end of line 46 of page 1.

Thanks for the suggestion. We explained earlier that NleE is a methyltransferase in the new manuscript.

Last paragraph of the introduction. The first sentence (page 3 line 70-71) is more or less a repeat of page 1 line 45/46. I would remove this sentence and substitute it with the last sentence of this paragraph.

We made the changes as the reviewer suggested.

Page 4 line 110. What is a "synthetase-tRNATyr suppression system"? either author state correctly that they use an orthodona amino acyl-tRNA synthetase(tRNA pair derived from the pair that incorporates Tyrosin in E. coli, or they just leave this detail, and may add a couple of words in the method section.

We made the changes as the reviewer suggested.

Page 5 line 141. SAM should be described already here, without waiting until page 6 lines 167/168. This is also a reason because NleE should be described before, otherwise the general reader does not understand what is SAM doing here.

We made the changes as the reviewer suggested.

I find the sentences from page 6 line 172 "our previous results… "until the end of the paragraph at page 7 line 176 more confusing than helpful. I recommend to remove these comments as they are unnecessary and not very clear.

Thanks for the suggestions. We deleted these sentences in the new version of manuscript.

Page 7 line 183. Before describing results, it should be explained that BetY was incorporated at several sites of NleE and BetY mutants were reacted with PSMD10. It should also be explained whether the Cys residues in PSMD10 are native or were inserted on purpose. Overall the paragraph between lines 180-191 of page 7 contains beautiful results but could be written in a clearer way.

We rewrote this part and explained that BetY was incorporated at several sites of NleE and BetY mutants were reacted with wild type PSMD10 which contains five Cys residues.